# RankMe: Assessing the Downstream Performance of Pretrained Self-Supervised Representations by Their Rank

## Abstract

Joint-Embedding Self Supervised Learning (JE-SSL) has seen a rapid development, with the emergence of many method variations and few principled guidelines that would help practitioners to successfully deploy those methods. The main reason for that pitfall actually comes from JE-SSL's core principle of not employing any input reconstruction. Without any visual clue, it becomes extremely cryptic to judge the quality of a learned representation without having access to a labelled dataset. We hope to correct those limitations by providing a single –theoretically motivated– criterion that reflects the quality of learned JE-SSL representations: their effective rank. Albeit simple and computationally friendly, this method —coined *RankMe*— allows one to assess the performance of JE-SSL representations, even on different downstream datasets, without requiring any labels, training or parameters to tune. Through thorough empirical experiments involving hundreds of repeated training episodes, we demonstrate how RankMe can be used for hyperparameter selection with nearly no loss in final performance compared to the current selection method that involves dataset labels. We hope that RankMe will facilitate the use of JE-SSL in domains with little or no labeled data.

## 1 Introduction

Self-supervised learning (SSL) has shown great progress to learn informative data representations in recent years (Chen et al., 2020a; He et al., 2020; Chen et al., 2020b; Grill et al., 2020; Lee et al., 2021; Caron et al., 2020; Zbontar et al., 2021; Bardes et al., 2021; Tomasev et al., 2022; Caron et al., 2021; Chen et al., 2021; Li et al., 2022a; Zhou et al., 2022a;b; HaoChen et al., 2021; He et al., 2022), catching up to supervised baselines and even surpassing them in few-shot learning, i.e., when evaluating the SSL model from only a few labeled examples. Although various SSL families of losses have emerged, most are variants of the joint-embedding (JE) framework with a siamese network architecture (Bromley et al., 1994), denoted as JE-SSL for short. The only technicality we ought to introduce to make our study precise is the fact that JE-SSL has introduced some different notations to denote an input's representation. In short, JE-SSL often composes a *backbone* or *encoder* network e.g., a ResNet50 and a *projector* network e.g., a multilayer perceptron. This projector is only employed during training, and we refer to its outputs as *embeddings*, while the actual inputs' *representation* employed for downstream tasks are obtained at the encoder's output.

Although downstream tasks performance of JE-SSL representations might seem impressive, one pondering fact should be noted: *all existing methods, hyperparameters, models — and thus performance — of JE-SSL are obtained by ad-hoc manual search involving the labels of the training samples*. In words, JE-SSL is tuned by monitoring the supervised performance of the model at hand. Hence, although labels are not directly employed to compute the weight updates, they are used as a proxy to signal the JE-SSL designer on how to refine their method. This single limitation prevents the deployment of JE-SSL in challenging domains where the number of available labelled examples is limited and such search can not be performed. Adding to the challenge, one milestone of JE-SSL is to move away from reconstruction based learning; hence without labels and without visual cues, tuning JE-SSL methods on unlabeled datasets remains challenging. This led to the application of feature inversion methods e.g. Deep Image Prior (Ulyanov et al., 2018) or conditional diffusion models (Bordes et al., 2021) to be deployed onto learned JE-SSL representation to try to visualize the

learned features. This first step towards removing the need for labels has seen some success but is doomed by the computational complexity of the methods, and their biases towards natural images i.e. it is not clear how such methods would perform on different data modalities.

In this study we propose RankMe, which is able to assess a model's performance without having access to any labels and without requiring any training or tuning. RankMe accurately predicts a model's performance both on In-Distribution (ID), i.e., same data distribution as used during the JE-SSL training, and on Out-Of-Distribution (OOD), i.e., different data distribution scenarios. We highlight this property at the top of fig. 1. The strength of RankMe lies in the fact that it is solely based on the singular values distribution of the learned embeddings, and thus does not rely on any parameters that need training, nor requires any ID/OOD labels. In fact, RankMe's motivation hinges on Cover's theorem (Cover, 1965) that states how increasing the rank of a linear classifier's input increases its training performance, and three simple hypotheses that we summarize below and thoroughly validate empirically. As such, RankMe provides a step towards (unlabeled) JE-SSL by allowing practitioners to cross-validate hyperparameters and select models without resorting to labels or feature inversion methods. We hope that RankMe will enable JE-SSL to be deployed even in challenging domains that possess no or little labelled data; we summarize our contributions below:

1. We introduce (eq. (1)) and motivate RankMe which combines Cover's theorem with the following three key hypotheses:
   - (H1) **increasing training performance increases testing performance** on both representations and embeddings i.e. no over-fitting is observed from the (non)linear probe, validated empirically in the bottom left of fig. 2
   - (H2) **embeddings' rank scale linearly between datasets.** Assuming a pretraining on the same dataset, if a set of embeddings has a greater rank than another on a dataset, it also holds on another one, validated empirically in the top row of fig. 2
   - (H3) **increasing embeddings performance increases representations performance**, validated empirically in the bottom right of fig. 2
   
   concluding that embeddings with greater rank will have greater train performance (Cover's theorem) and test performance (H1) on ID and OOD cases (H2), even before the projector (H3).
2. We demonstrate that RankMe's ability to assess JE-SSL downstream performance is robust across methods, e.g. VICReg, SimCLR, and their variants, and is also robust to architecture changes, e.g. using a projector network and/or a nonlinear evaluation method (see fig. 3 and section 3.3).
3. We demonstrate that RankMe enables hyperparameter cross-validation for any given JE-SSL method; RankMe is able to retrieve and sometimes surpass most of the performance previously found by manual search using labels, on both in domain and out of domain datasets; see bottom of fig. 1 and table 1.

We provide a hyperparameter free numerically stable implementation of RankMe in section 3.1 and pseudo-code for cross-validation in fig. 5. Through extensive experiments involving 11 different datasets and more than 85 trained models over 4 methods, we demonstrate that in the linear and nonlinear probing regime, RankMe is able to tell apart high and low performing models, even on different downstream tasks without having access to labels or downstream task data samples.

## 2 RELATED WORKS

**Joint embedding self-supervised learning (JE-SSL).** In JE-SSL, two main families of method can be distinguished: contrastive and non-contrastive. Contrastive methods (Chen et al., 2020a; He et al., 2020; Chen et al., 2020b; 2021; Yeh et al., 2021) mostly rely on the InfoNCE criterion (Oord et al., 2018) except for HaoChen et al. (2021) which uses squared similarities between the embedding. A clustering variant of contrastive learning has also emerged (Caron et al., 2018; 2020; 2021) and can be thought of as contrastive methods, but between cluster centroids instead of samples. Non-contrastive methods (Grill et al., 2020; Chen & He, 2020; Caron et al., 2021; Bardes et al., 2021; Zbontar et al., 2021; Ermolov et al., 2021; Li et al., 2022b) aim at bringing together embeddings of positive samples, similar to contrastive learning. However, a key difference with contrastive learning lies in how those methods prevent a representational collapse. In the former, the criterion explicitly pushes away negative samples, i.e., all samples that are not positive, from each other. In the latter, the criterion does not prevent collapse by distinguishing positive and negative samples, but instead considers the embeddings as a whole and encourages information content maximization e.g., by regularizing

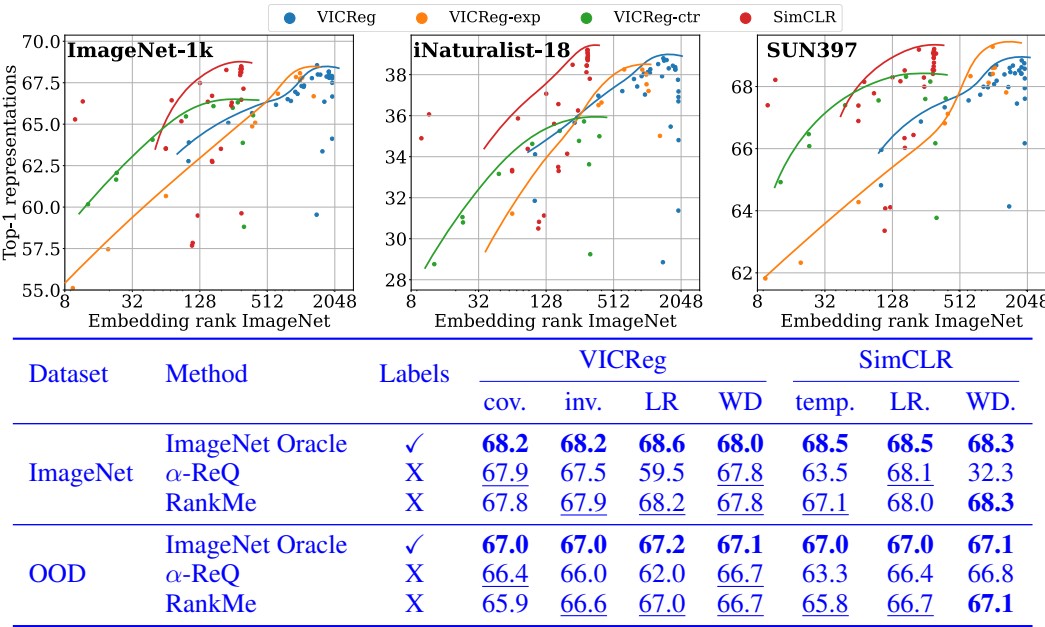

| Dataset | Method | Labels | VICReg | | | | SimCLR | | |
|---|---|---|---|---|---|---|---|---|---|
| | | | cov. | inv. | LR | WD | temp. | LR. | WD. |
| ImageNet | ImageNet Oracle | ✓ | **68.2** | **68.2** | **68.6** | **68.0** | **68.5** | **68.5** | **68.3** |
| | $\alpha$-ReQ | X | 67.9 | 67.5 | 59.5 | 67.8 | 63.5 | 68.1 | 32.3 |
| | RankMe | X | 67.8 | 67.9 | 68.2 | 67.8 | 67.1 | 68.0 | **68.3** |
| OOD | ImageNet Oracle | ✓ | **67.0** | **67.0** | **67.2** | **67.1** | **67.0** | **67.0** | **67.1** |
| | $\alpha$-ReQ | X | 66.4 | 66.0 | 62.0 | 66.7 | 63.3 | 66.4 | 66.8 |
| | RankMe | X | 65.9 | 66.6 | 67.0 | 66.7 | 65.8 | 66.7 | **67.1** |

Figure 1: **(Top)** Performance of JE-SSL representations (encoder output) in **y-axis** against the embeddings (projector output) RankMe values in **x-axis** on ImageNet-1k. Except for some degenerate solutions at full-rank, RankMe values correlate well with in-distribution (**left column**) and out-of-distribution (**right columns**) classification performance. **(Bottom)** Hyperparameter selection using the common supervised linear probe strategy and the proposed unsupervised RankMe strategy. OOD indicates the average performance over all the considered dataset other than ImageNet. Without any label, optimization or parameters, RankMe is able to recover most of the performance obtained by using ImageNet validation set, highlighting its strength as a hyperparameter selection tool. RankMe also outperforms $\alpha$-ReQ on average and does not suffer from as big performance drops in worst cases.

the empirical covariance matrix of the embeddings. Such a categorization is not needed for our development, and we thus refer to any of the above method as JE-SSL.

**Dimensional collapse in JE-SSL.** The phenomenon of learning rank-deficient embeddings, or dimensional collapse, in JE-SSL has recently been studied from both a theoretical and empirical point of view. The empirical emergence of dimensional collapse was studied in Hua et al. (2021) where they proposed the use of a whitening batch normalization layer to help alleviate it. In Jing et al. (2022), a focus on contrastive approaches in a linear setting enabled a better understanding of dimensional collapse and the role of augmentations in its emergence. Performance in a low label regime of a partially collapsed encoder can also be improved by forcing the whitening of its output, as shown in He & Ozay (2022). Furthermore, it was shown in Balestriero & LeCun (2022) how dimensional collapse is a phenomenon that should not necessarily happen in theory and how its emergence is mostly due to practical concerns. Interestingly, we will see through the lens of RankMe that while reducing dimensional collapse is often beneficial, doing so "at all cost" can lead to degenerate solutions. The collapse induced from training using a softmax layer was also studied in Ganea et al. (2019), where they show that high rank embeddings are desirable.

**Evaluation of JE-SSL representations.** Evaluating the representations learned by JE-SSL methods is fundamental to enable the optimal selection of those methods' hyperparameters, which are numerous. Yet, due to the imprecise nature of what makes a good representation, multiple strategies have emerged which evaluate different properties of representations. The most common approach relies on the *strong assumption* of having labels on the dataset where the JE-SSL method is trained on. In this case, on trains a linear classifier on the JE-SSL representations (Misra & Maaten, 2020) and directly use the test accuracy to compare models. This method was extended to the use of nonlinear classifiers, e.g., a $k$-nn classifier (Wu et al., 2018; Zhuang et al., 2019). Performance evaluation without labels can also be done using a pretext-task, such as rotation prediction. This technique helped in selecting data augmentation policies in Reed et al. (2021). One limitation lies in the need to select and train the classifier of the pretext-task, and on the strong assumption that rotation were

not part of the transformations one aimed to be invariant to. Since (supervised) linear evaluation is the most widely used evaluation method, we will focus on showing how RankMe compares with it. Most related to us is Ghosh et al. (2022) where representations are evaluated by their eigenspectrum decay, giving a baseline for unsupervised hyperparameter selection.

## 3 REPRESENTATIONS' RANK CORRELATE WITH DOWNSTREAM PERFORMANCE ACROSS TASKS AND MODELS

The goal of this section is to formally introduce and motivate RankMe while providing a numerically stable implementation (section 3.1). The construction of RankMe hinges on three hypotheses that we validate empirically throughout this section.

### 3.1 RANKME: FROM THEORY TO IMPLEMENTATION

We first want to build notations and intuition into the construction of RankMe. To that hand, we first quantify approximation and classification errors of learned embeddings as a function of their rank, and then motivate how embeddings' rank can be sufficient to compare test performance of JE-SSL models's representations. This criterion should however only be used to compare different runs of a given method, since the embeddings' rank is not the only factor that affects performance. To ease notations, we refer to the (train) dataset used to obtain the JE-SSL model as the *source dataset*, and the test set on the same dataset or a different OOD dataset as the *target dataset*.

**From Source Embeddings's Rank to Target Representations performance.** We first build some intuition in the regression settings. In this case, a common linear algebra result ties the best-case and worst-case approximation error of any target matrix $\boldsymbol{Y} \in \mathbb{R}^{N \times C}$ from a rank-$R$ matrix $\boldsymbol{P} \in \mathbb{R}^{N \times C}$ to the singular values of $\boldsymbol{Y}$ that run from $R$ to the rank of $\boldsymbol{Y}$ when ordered in decreasing order. Without loss of generality, we only consider the case $C > N$ in this study, i.e., we have more samples than dimensions. Formally, this provides a lower bound on

$$\|\boldsymbol{Y} - \boldsymbol{P}\|_F^2 \geq \sum_{r=R+1}^{C} \sigma_r^2(\boldsymbol{Y}),$$

which is tight for $\boldsymbol{P}$ of rank $R$, and with $\sigma_k$ the operator returning the $k^{\text{th}}$ singular value of its argument, ordered in decreasing order. This result, on which RankMe relies on, demonstrates that a necessary (but not sufficient) condition for an approximation $\boldsymbol{P}$ to well approximate $\boldsymbol{Y}$ is to have at least the same rank as $\boldsymbol{Y}$. A similar result can be obtained in classification by considering multiple one-vs-all classifiers. In practice, however, we commonly employ a linear probe network on top of given embeddings $\boldsymbol{Z}$ to best adapt them to the target $\boldsymbol{Y}$, i.e., $\boldsymbol{P} = \boldsymbol{ZW} + \boldsymbol{1b}^T$. However, a linear transformation is not able to increase the rank of the input matrix since

$$\text{rank}(\boldsymbol{P}) \leq \min(\text{rank}(\boldsymbol{Z}), \text{rank}(\boldsymbol{W})) + 1.$$

We directly obtain that $\min_{\boldsymbol{W}, \boldsymbol{b}} \|\boldsymbol{Y} - \boldsymbol{ZW} - \boldsymbol{1b}^T\|_F^2 \geq \sum_{r=R+1}^{C} \sigma_r^2(\boldsymbol{Y})$. In short, the approximation lower bound is not improved by allowing linear transformation of the embeddings. Further supporting the above, we ought to recall Cover's theorem (Cover, 1965) stating that the probability of a randomly labeled set of points being linearly separable only increases if $N$ is reduced or $R$ is increased. We formalize those results below.

**Proposition 1.** *The maximum training accuracy of given embeddings in linear regression or classification increases with their rank. For classification, it plateaus when the rank surpasses the number of classes.*

We thus introduce RankMe formally as the following smooth rank measure, originally introduced in Roy & Vetterli (2007),

$$\text{RankMe}(\boldsymbol{Z}) = \exp\left(-\sum_{k=1}^{\min(N,K)} p_k \log p_k\right), \ p_k = \frac{\sigma_k(\boldsymbol{Z})}{\|\sigma(\boldsymbol{Z})\|_1} + \epsilon, \tag{1}$$

where $\boldsymbol{Z}$ is the source dataset's embeddings. By noticing that RankMe provides a smooth measure of the embeddings' rank (more details in the implementation section) we can lean on proposition 1 to

see that given two models, the one with greater RankMe value will have greater training performance. This is only guaranteed for different models of the same method, since embedding rank is not necessarily the only factor that affects performance.

The above result is however not too practical yet since what we are truly interested in are (i) performance on unseen samples, i.e., on the test set and out-of-distribution tasks, and (ii) performance on the representations and not the embeddings since it is common to ablate the projector network of JE-SSL models. Below, we validate three key hypotheses which, when verified, imply that we can extend the impact of RankMe such that *(OOD) test performance of JE-SSL representations are increased when RankMe's value on their train set embeddings is increased.*

**Validating RankMe's Hypotheses** The development of RankMe is theoretically grounded when it comes to guaranteeing improved source dataset embeddings performance. To empirically extend it to target dataset representations performance we need to verify three hypotheses: (i) linear probes do not overfit, (ii) embeddings and representations performance are monotonically linked, and (iii) source and (OOD) target embeddings ranks are monotonically linked. Due to the different nature of datasets used for downstream tasks, there is no inherent reason for the rank of embeddings to transfer in a monotonic way to them. However, if the source dataset is diverse enough and target datasets have some semantic overlap with the source dataset, then we have

$$\text{rank}(\boldsymbol{Z}_{\text{target}}) \propto \text{rank}(\boldsymbol{Z}_{\text{source}}). \tag{2}$$

We observe in section 3.2 and fig. 2 that the rank of JE-SSL representations scales linearly between different input distributions e.g. going from a *source* task such as Imagenet (Deng et al., 2009) to a *target* task such as iNaturalist. This is further confirmed by Pearson correlation coefficients greater than $0.99$, except for StanfordCars where it is $0.88$. Interestingly, we observe that the StanfordCars dataset suffers from a less distinctive linear scaling due to the dataset distribution having a small overlap with ImageNet. This indicates that as long as the source dataset is relatively diverse, then using RankMe to select a model with greater embeddings' rank will also correspond to selecting a model with greater embeddings' rank on the target dataset.

Furthermore, as the train performance increases, so does the test performance. We validate this in fig. 2. As a result, using RankMe to select a model with greater train performance is enough to also select a model with greater test performance.

Finally, we report in fig. 2 that the performance of embeddings and representations scales almost monotonically. These results are supported by visualization of representations and embeddings from feature inversion models (Bordes et al., 2021). Hence, using RankMe to select the model maximizing the performance on the former also selects a model maximizing performance on the latter.

With these three hypotheses validated empirically, we can confidently say that RankMe computed on the embeddings of the source dataset is a predictor of representations' performance on target datasets.

**Robust RankMe Implementation.** One of the most crucial step of RankMe is the estimation of the embeddings' rank. A trivial solution could be to check at the number of nonzero singular values. Denoting by $\sigma_k$ the $k^{\text{th}}$ singular value of the $(N \times K)$ embedding matrix $\boldsymbol{Z}$, this would lead to $\text{rank}(\boldsymbol{Z}) = \sum_{k=1}^{\min(N,K)} 1_{\{\sigma_k > 0\}}$. However, such a definition is too rigid for practical scenarios. For example, round-off error alone could have a dramatic impact on the rank estimate. Instead, alternative and robust rank definitions have emerged (Press et al., 2007) such as $\text{rank}(\boldsymbol{Z}) = \sum_{k=1}^{\min(N,K)} 1_{\{\sigma_k > \max_i \sigma_i \times \max(M,N) \times \epsilon\}}$, where $\epsilon$ is a small constant dependent on the data type, typically $10^{-7}$ for `float32`.

An alternative measure of rank comes from a probability viewpoint where the singular values are normalized to sum to 1 and the Shannon Entropy (Shannon, 1948) is used, which corresponds to our definition of RankMe from eq. (1). As opposed to the classical rank, the chosen eq. (1) does not rely on specifying the exact threshold at which the singular value is treated as nonzero. Throughout our study, we employ eq. (1), and provide the matching analysis with the classical rank in the appendix. Another benefit of RankMe's eq. (1) is in its quantification of the whitening of the embeddings in addition to their rank, which is known to simplify optimization of (non)linear probes put on top of them (Santurkar et al., 2018). Lastly, although eq. (1) is defined with the full embedding matrix $\boldsymbol{Z}$, we observe that not all the samples need to be used to have an accurate estimate of RankMe. In practice, we use 25600 samples as ablation studies provided in appendix G and fig. S11 indicate that this provides a highly accurate estimate.

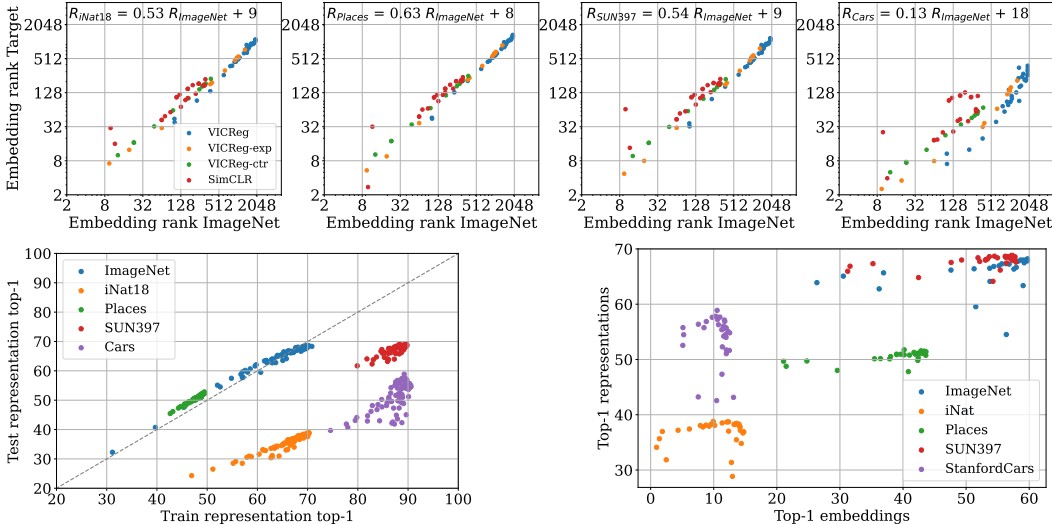

Figure 2: Validation of the hypotheses motivating RankMe.**(Top)** Embeddings' rank transfers from source to target datasets. The estimates use 25600 images from the respective datasets.**(Bottom Left)** Train and test accuracy are highly correlated across datasets.**(Bottom Right)** An increase in performance on embeddings leads to an increase in performance on representations.

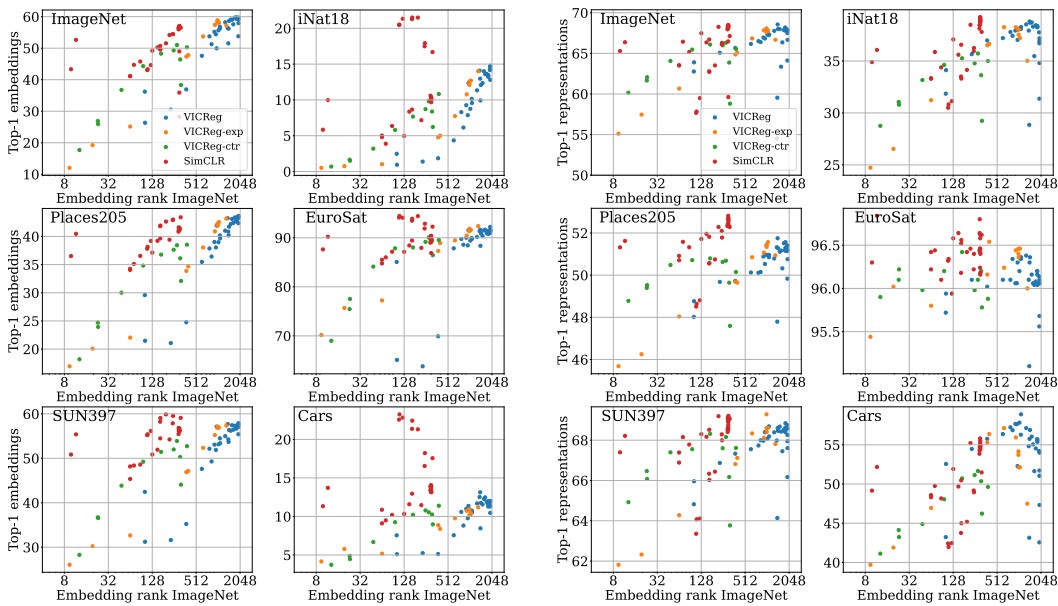

Figure 3: **(Left)** Validation of RankMe on embeddings, a higher ImageNet rank leads to improved performance across methods and datasets.**(Right)** Validation of RankMe on representations, where the link is even clearer, reinforcing RankMe's practical use.

## 3.2 RANKME PREDICTS LINEAR PROBING PERFORMANCE EVEN ON UNSEEN DATASETS

In order to empirically validate RankMe, we compare it to linear evaluation, which is the default evaluation method of JE-SSL methods. Finetuning has gained in popularity with Masked Image Modeling methods, but this can have a significant impact on the properties of the embeddings and alters what was learned during the pretraining. As such, we do not focus on this evaluation.

**Experimental Methods and Datasets Considered** In order to provide a meaningful assessment of the embeddings rank's impact on performance, we focus on 4 JE-SSL methods. We use SimCLR as a representative contrastive method, VICReg as a representative covariance based method, and

VICReg-exp and VICReg-ctr which were introduced in Garrido et al. (2022). To make our work self-contained, we present the methods in appendix A. We chose to use VICReg-exp and VICReg-ctr as they provide small modifications to VICReg and SimCLR while producing embeddings with different rank properties. For each method we vary parameters that directly influence the rank of the embeddings, whether it is the temperature use in softmax based methods, which directly impacts the hardness of the softmax, or the loss weights to give more or less importance to the regularizing aspect of loss functions. We also vary optimization parameters such as the learning rate and weight decay to provide a more complete analysis. We provide the hyperparameters used for all experiments in appendix K. All approaches were trained in the same experimental setting with a ResNet-50 (He et al., 2016) backbone with a MLP projector having intermediate layers of size $8192, 8192, 2048$, which avoids any architectural rank constraints. The models were trained for 100 epochs on ImageNet with the LARS (You et al., 2017; Goyal et al., 2017) optimizer.

In order to evaluate the methods, we used ImageNet (our source dataset), as well as iNaturalist18 (Horn et al., 2018), Places205 (Zhou et al., 2014), EuroSat (Helber et al., 2019), SUN397 (Xiao et al., 2010) and StanfordCars (Krause et al., 2013) to evaluate the trained models on unseen datasets. These commonly used datasets provide a wide range of scenarios that differ from ImageNet and provide meaningful ways to test the robustness of RankMe. For example, iNaturalist18 consists of 8412 classes focused on fauna and flora which requires more granularity than similar classes on ImageNet, SUN397 focuses on scene understanding, deviating from the single object and object-centric images of ImageNet, and EuroSat consists of satellite images which again differ from ImageNet. Datasets such as iNaturalist can allow theoretical limitations to manifest themselves more clearly due to the number of classes being significantly higher than the rank of learned representations. While we focus on these datasets for our visualizations, we also include CIFAR10, CIFAR100 Krizhevsky et al. (2009), Food101 Bossard et al. (2014), VOC07 Everingham et al. and CLVR-count Johnson et al. (2017) for our hyperparameter selection results, and provide visualizations in appendix D. In order to evaluate on those datasets, we relied on the VISSL library (Goyal et al., 2021). We provide complete details on the pretrainings and evaluations in appendix I.

**RankMe as a prediction of linear classification accuracy.** As can be seen in fig. 3, for a given method the performance on the embedding is improved by with a higher embedding rank, whether we look on ImageNet on which the models were pretrained or on downstream datasets. Nonetheless, there are some visible outliers, but they are mostly on SimCLR in settings with very high error rates compared to before the projector, such as in iNaturalist or StanfordCars. The conceptual closeness between VICReg-ctr and SimCLR pointed out in Garrido et al. (2022) would also suggest that these results need to be interpreted carefully, but they do reinforce the fact that a high rank is a necessary and not sufficient condition for improved performance on downstream tasks. It is also very tempting to draw conclusions when comparing different approaches, especially when looking at the ImageNet performance, however since dimensional collapse is not the only performance deciding factor one should refrain from doing so. The link between embedding rank and performance is even clearer when evaluating on the representations, as is usually done. In this scenario the link is more consistent across datasets, where we observe again that a higher rank is necessary for improved performance. This solidifies the use of RankMe as a performance metric that can be used in practice.

## 3.3 Going Further: RankMe Also Holds for Nonlinear Probing and for Different Architectures

**Non-linear evaluation.** While we have been focusing on linear evaluation, one can wonder if the behaviour changes when using a more complex task-related head. We thus give some evidence that the previously observed behaviours are similar with a non-linear classification head. We used a simple 3 layer MLP with intermediate dimensions $2048$, where each layer is followed by a ReLU activation. This choice of dimensions ensures that there are no architectural rank constraints on the embeddings. We focused on SUN397 and StanfordCars for this study due to their conceptual differences to ImageNet. The low rank of embeddings produced by SimCLR on these datasets would suggest that a non-linear classifier might help improve performance, since it is not as theoretically limited by the embeddings' rank as it is in the linear setting. However, as we can see in fig. 4, the behaviors for all methods is the same as in the linear regime. This would suggest that RankMe is also a suitable metric to evaluate downstream performance in a non-linear setting.

**Dimensional collapse on different architectures.** Our results so far have only focused on ResNet-50s, and a concern could be that the architecture played a significant role in the introduction of

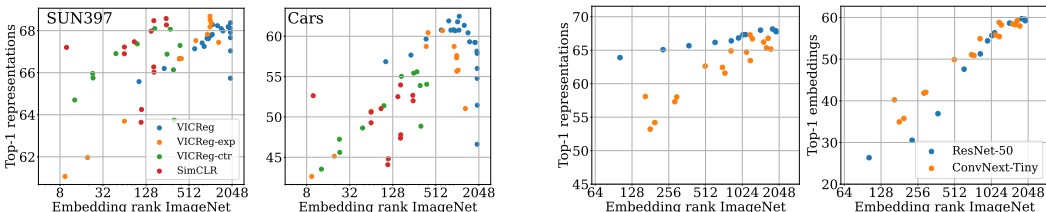

Figure 4: Impact of rank on performance on other architectures and evaluation protocols. **(Left)** Using a 3 layer MLP as classification head does not alter the performance before or after the projector, showing that RankMe can go beyond linear evaluation.**(Right)** ConvNexts are also sensitive to dimensional collapse, showing that rank-deficiency is not an artifact of ResNets.

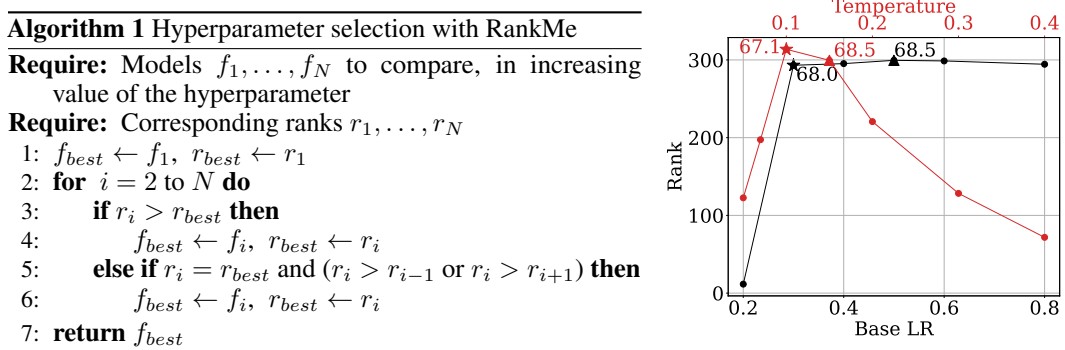

**Algorithm 1** Hyperparameter selection with RankMe
___
**Require:** Models $f_1, \ldots, f_N$ to compare, in increasing value of the hyperparameter
**Require:** Corresponding ranks $r_1, \ldots, r_N$
1: $f_{best} \leftarrow f_1,\ r_{best} \leftarrow r_1$
2: **for** $i = 2$ to $N$ **do**
3:     **if** $r_i > r_{best}$ **then**
4:         $f_{best} \leftarrow f_i,\ r_{best} \leftarrow r_i$
5:     **else if** $r_i = r_{best}$ and ($r_i > r_{i-1}$ or $r_i > r_{i+1}$) **then**
6:         $f_{best} \leftarrow f_i,\ r_{best} \leftarrow r_i$
7: **return** $f_{best}$

Figure 5: **(Left)** Algorithm describing how to use RankMe for hyperparameter selection. We select either the highest rank model, or if there are multiple ones, the one with the minimal/maximal value achieving it.**(Right)** Visual example of the hyperparameter selection applied to SimCLR's temperature and learning rate. The star indicates the value that is selected using RankMe, and the triangle the one with the ImageNet oracle. Notice the high rank of oracle selected models.

collapse. As such, we trained VICReg in the same setting as before but using ConvNext-T (Liu et al., 2022) as the backbone architecture. As we can see in fig. 4, collapse still appears in this case, with an even stronger impact on performance on ImageNet. This reinforces the findings of Jing et al. (2022); He & Ozay (2022), which study collapse through the used loss function independently of the architecture of the backbone.

**RankMe in more diverse settings.** While our focus has been on contrastive methods, we further study in appendix C how RankMe can be applied to clustering methods such as DINO, where it shows great effectiveness. We also take a look at the effectiveness of RankMe when pretraining on other source datasets in appendix B, validating RankMe on iNaturalist18 pretraining.

## 4 RANKME FOR LABEL-FREE HYPERPARAMETER SELECTION IN SSL

We previously focused on validating RankMe by comparing overall performance compared to linear evaluation. In this section we focus on the evolution of rank and performance when varying one hyperparameter at a time in order to demonstrate how RankMe can be used for hyperparameter selection. We focus on loss specific hyperparameters such as the loss weights or temperature as well as hyperparameters related to optimization, such as the learning rate and weight decay.

### 4.1 USING RANKME TO CHOOSE THE CORRECT HYPERPARAMETER VALUE

As we have shown before, having a higher rank is necessary for better performance, and using RankMe to find the best value of an hyperparameter is as simple as choosing the value that leads to the highest rank, as illustrated in fig. 5. Certain hyperparameters will lead to plateaus of equal rank, and in those the value that first achieves the maximal value should be selected. This second part is however only applicable when hyperparameter values can be ordered.

Even in cases where the values cannot be compared, and equal ranks are found in a different setting, this still makes it possible to discard some runs and only focus on the one that achieve the maximal

Table 1: Top-1 accuracies obtained by doing hyperparameter selection using ImageNet validation performance, $\alpha$-ReQ or RankMe. OOD indicates the average performance over all the considered datasets other than ImageNet. The performance is computed on the embeddings.

| Dataset | Method | VICReg | | | | SimCLR | | |
|---------|--------|--------|--------|------|------|--------|------|------|
| | | cov. | inv. | LR | WD | temp. | LR. | WD. |
| ImageNet | ImageNet Oracle | **59.7** | **59.7** | **59.7** | **59.7** | **56.9** | **56.9** | **57.1** |
| | $\alpha$-ReQ | 59.6 | 59.2 | 36.2 | 59.3 | 51.5 | 56.4 | 49.0 |
| | RankMe | 59.6 | **59.7** | **59.7** | 59.5 | 56.5 | 56.0 | **57.1** |
| OOD | ImageNet Oracle | 52.1 | **52.5** | 52.1 | **52.3** | 51.4 | **51.4** | 51.5 |
| | $\alpha$-ReQ | **52.3** | **52.5** | 44.2 | 51.9 | **54.2** | 51.3 | 51.5 |
| | RankMe | **52.3** | **52.5** | 52.1 | 51.8 | 53.4 | 51.1 | 51.5 |

rank. This further highlight how maximal rank is only a necessary condition for good performance. Nonetheless, when the hyperparameters are ordered we can go one step further and use the rank alone to find a good hyperparameter value.

## 4.2 EXPERIMENTS

In order to demonstrate the effectiveness of RankMe for hyperparameter selection, we apply the algorithm presented in fig. 5 to find the best values for a given set of hyperparameters for VICReg and SimCLR. Our focus is on the covariance and invariance weights in VICReg, the temperature in SimCLR, and on learning rate and weight decay for both. We compare the performance on ImageNet as well as the average performance on the previously discussed OOD datasets to models selected by their ImageNet top-1 accuracy on its validation set. For per dataset performance, confer appendix J. As can be seen in table 1, using RankMe we are able to retrieve most of the performance on ImageNet, with gaps being lower than half a point. It is not possible to beat the selection using ImageNet's validation, since this is the metric we are evaluating on. However, on OOD datasets we are able to improve the performance in certain settings, and match it in the others. Thus, when comparing performance after the projector, RankMe is the better approach of the two to select the hyperparameters that will generalize best to unseen datasets. When comparing to $\alpha$-Req, RankMe achieves better in domain performance, but on OOD datasets $\alpha$-ReQ performs slightly better, though with bigger worst case performance drops. We provide an in-depth analysis of $\alpha$-ReQ in appendix E, where we find that the power law prior of $\alpha$-ReQ fails on the embeddings and as such those results must be interpreted with care. As pointed out in Girish et al. (2022), using ImageNet performance to select models can lead to suboptimal performance in downstream tasks, which our results further confirm and reinforces the need for a new way of selecting hyperparameters. When looking at performance before the projector in fig. 1, we can see that here RankMe does not beat the models selected with ImageNet's validation set, even on OOD datasets. However, RankMe performs better than $\alpha$-ReQ in most settings, while not suffering from severe drops in the worst cases. Nevertheless, the gaps between RankMe and the ImageNet oracle are on average of less than half a point, which shows how competitive RankMe can be for hyperparameter selection, despite using no labeled data, having no parameters to tune, and being able to be computed in a couple of minutes.

## 5 CONCLUSION

We have shown how the phenomenon of dimensional collapse in self-supervised learning can be used as a powerful metric to evaluate models. By using a theoretically motivated analogue of the rank of embeddings, we show that the performance on downstream datasets can easily be assessed by only looking at the training dataset, without any labels, training, or parameters. While our work focuses on linear classification, we show promising results in non-linear classification that raise the question of how general this simple metric can be. Furthermore, its competitiveness with traditional oracle based hyperparameter selection methods makes it a promising tool in settings where labels are scarce, such as in the case of large uncurated datasets. As such, this work makes a step towards completely label-less self-supervised learning, as most existing approaches' hyperparameters are tuned with the help of ImageNet's validation set. Further work will explore the use of RankMe in more varied scenarios, to further legitimize its use in designing better self-supervised approaches.

## 6 REPRODUCIBILITY STATEMENT

While reproducing the pretrainings is prohibitively expensive since each training takes around a day on 8 V100 GPUs, we provide all of the hyperparameters used in appendix K. We also provide all of the pretraining details in appendix I, along with the hyperparameters used for the linear evaluations in the same section. We also provide all the performance and rank values to reproduce our main figures in appendix K. While the implementation of RankMe is straightforward, we provide an example algorithm to use it in fig. 5. All of these efforts should make our results reproducible and verifiable.

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

## A BACKGROUND

In order to make our work as self-contained as possible, we recall the loss functions of the methods we study. For conciseness, we refer to the outputs of the encoder as *representations* and the outputs of the projection head as *embeddings*, which we denote by $z_i \in \mathbb{R}^d$. We first briefly recall that the SimCLR loss is given by

$$\mathcal{L} = - \sum_{(i,j)\in\mathbb{P}} \frac{e^{CoSim(\boldsymbol{z}_i,\boldsymbol{z}_j)}}{\sum_{k=1}^{N} \mathbf{1}_{\{k\neq i\}} e^{CoSim(\boldsymbol{z}_i,\boldsymbol{z}_k)}},$$

with $\mathbb{P}$ the set of all positive pairs in the current mini-batch or dataset that comprise $N$ exemplars.

VICReg's loss is defined with three components. The variances loss $v$ acts as a norm regularizer for the dimensions, and the covariance loss aims at decorrelating dimensions in the embeddings. They are respectively defined as

$$v(\boldsymbol{Z}) = \frac{1}{d}\sum_{i=1}^{d} \max\left(0, 1 - \sqrt{\mathrm{Var}(Z_{\cdot,i})}\right) \text{ and } c(\boldsymbol{Z}) = \frac{1}{d}\sum_{i\neq j} \mathrm{Cov}(\boldsymbol{Z})^2.$$

Both of these loss are combined with an invariance loss that matches positives pairs, giving a final loss of

$$\mathcal{L} = \lambda \sum_{(i,j)\in\mathbb{P}} \|z_i - z_j\|_2^2 + \mu\, c(\boldsymbol{Z}) + \nu\, v(\boldsymbol{Z}).$$

VICReg-exp is defined similarly, but with the exponential covariance loss defined as

$$c_{exp}(\boldsymbol{Z}) = \frac{1}{d}\sum_i \log\left(\sum_{j\neq i} e^{\mathrm{Cov}(\boldsymbol{Z})_{i,j}/\tau}\right). \tag{3}$$

VICReg-ctr is then VICReg-exp but applied to $\boldsymbol{Z}^T$, making it a contrastive approach and conceptually similar to SimCLR. These methods give us different scenarios of collapse and allow us to make a more general study of the rank of representations as a powerful metric.

## B INATURALIST18 PRETRAINING

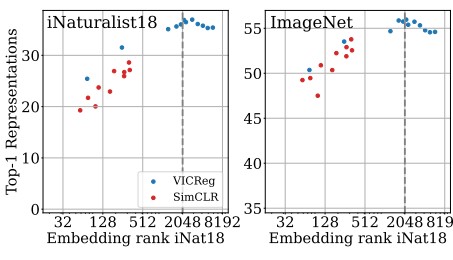

| Dataset | Method | SimCLR | VICReg |
|---|---|---|---|
| | | temp. | cov. |
| iNat18 | iNat18 Oracle | 28.6 | 37.0 |
| | RankMe | 27.1 | 36.9 |
| | Difference | −1.5 | −0.1 |
| ImageNet | iNat18 Oracle | 53.8 | 55.6 |
| | RankMe | 52.5 | 55.9 |
| | Difference | −1.3 | +0.3 |

Figure S1: RankMe applied to iNaturalist18 pretrainings **(Left)**. RankMe is able to select hyperparameters when pretraining on iNaturalist **(Right)**.

While our experiments have previously focused on ImageNet pretraining due to its wide use in the community, one can wonder if RankMe is still applicable when trained on another source dataset. To verify this, we pretrained VICReg and SimCLR on iNaturalist18, and respectively varied the covariance loss' weight and the temperature to study the influence on the rank of embeddings. We used the same protocol as ImageNet pretraining but trained for 300 epochs instead of 100 to obtain a similar number of iterations. We then evaluated the performance on iNaturalist18 and ImageNet. We use 8192 dimensional embeddings due to the high number of classes of iNaturalist, but since the representations are 2048 dimensional, the rank cannot intrinsically go higher so we consider that all highers ranks are effectively 2048.

As we can see in fig. S1, RankMe provides a similar level of performance as on ImageNet pretrainings, validating it on a different source dataset. RankMe is even able to improve performance on ImageNet compared to the iNaturalist18 oracle, further showing the limitations of such oracles on downstream tasks.

## C  APPLICABILITY TO CLUSTER BASED METHODS

While we have studied the applicability of RankMe on contrastive methods, cluster based methods such as DINO have become extremely popular, and since the definition of embeddings is not as clear cut in them, a thorough analysis is required. We will proceed in two steps:

- Show that dimensional collapse happens right before the clustering layer, and not on the prototypes
- Show that RankMe is a good measure of performance on DINO

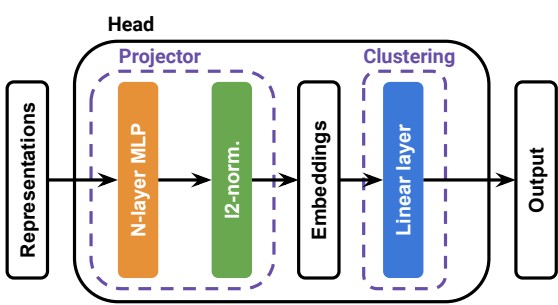
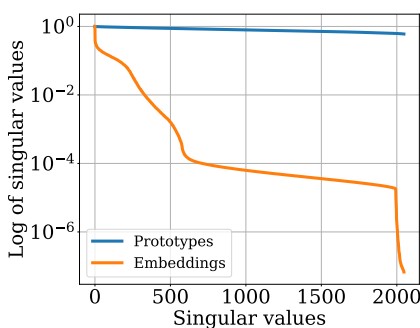

Figure S2: DINO's projection head can be split in two parts, a classical projector and a clustering layer (**Left**). Collapse happens before the clustering layer and not on the clustering prototypes (**Right**).

As we can see in figure S2, DINO's projector can be interpreted as both a classical projector and a clustering layer, whose weights are clustering prototypes. This interpretation comes from the softmax that is applied on the output of the projection head which can be interpreted as an InfoNCE between the embeddings and the clustering prototypes that make up the clustering layer. While prototypes themselves are not particularly collapsed, the embeddings that are obtained before the clustering present dimensional collapse.

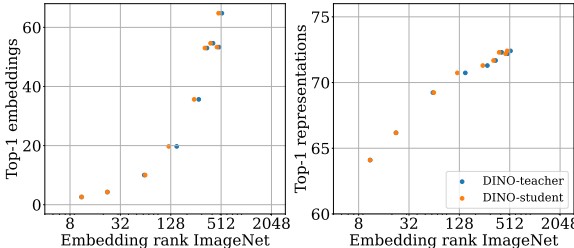

| Method | Temperature | |
|---|---|---|
| | Student | Teacher |
| ImageNet Oracle | 72.4 | 72.3 |
| RankMe | 72.4 | 72.2 |
| Difference | = | −0.1 |

Figure S3: RankMe is able to measure DINO's performance on its source dataset (**Left**). DINO's hyperparameters can be selected by using RankMe (**Right**).

As we can see in fig. S3, the phenomenon of dimensional collapse is highly visible in DINO, which enables the use RankMe to find optimal hyperparameter values. Validating this on ImageNet, we see that RankMe is able to match the performance of the oracle, or leads to slightly lower performance, further validating RankMe on another popular method.

## D  RESULTS ON SUPPLEMENTARY DATASETS

While we previously focused on certain datasets for their interesting natures, we will provide additional visualizations for the remaining datasets.

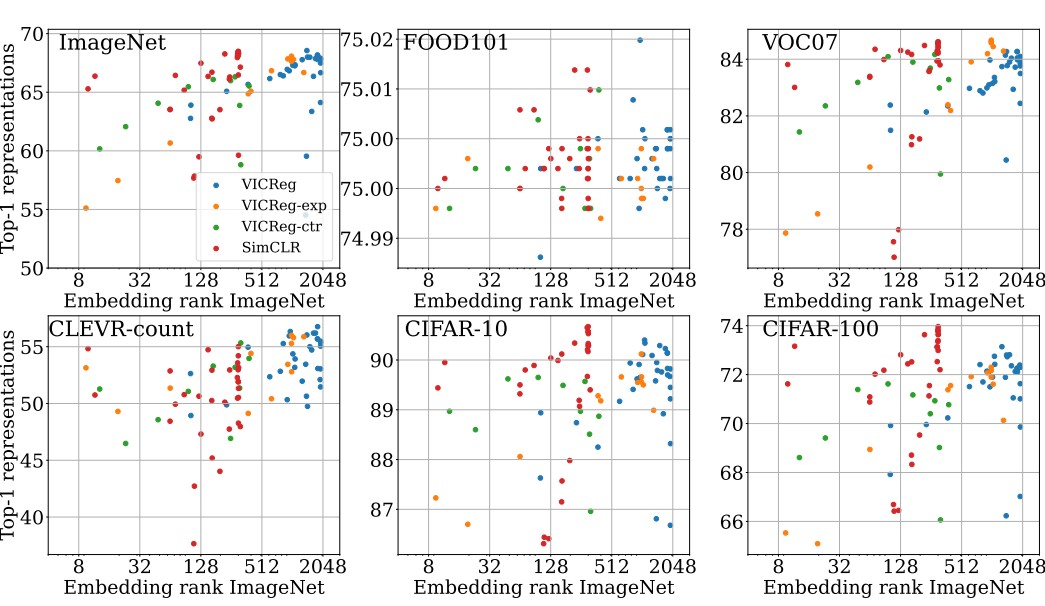

Figure S4: Link between embedding rank and downstream performance on the embeddings.

Figure S5: Link between embedding rank and downstream performance on the representations.

As we can see in figs. S4 and S5, we find similar behaviours as before, apart from Food101 where performance are almost identical for all methods. This reinforces the previous validation of RankMe. The relative simplicity of the datasets targeted here make the theoretical limitations of rank-deficient embeddings harder to see, even though we still see that a high rank helps generalization.

# E    DETAILED RESULTS FOR $\alpha$-REQ

In order to further study the performance of $\alpha$-ReQ, we reproduce our plots for RankMe using $\alpha$-ReQ instead of the rank of embeddings. We compare both the intended use of $\alpha$-ReQ in fig. S6, as well as applying it on the embeddings to measure performance on the representations, which we foudn was necessary for RankMe in fig. S7.

As we can see in fig. S6, there are no clear link visible between the value of $\alpha$-ReQ and downstream performance. Especially we are unable to see the tendency of performance to increase as $\alpha$ tends to one. Nonetheless $\alpha$-ReQ was still able to lead to good performance when used for hyperparameter selection.

When applying $\alpha$-ReQ as we would RankMe, we can see in fig. S7 that there is again no trend of performance increase when $\alpha$ tends to one. On the contrary we even find that performance tends to get better with a lower $\alpha$, as is most visible in StanfordCars, iNaturalist18 or ImageNet for example. $\alpha$ going towards one means that the singular values of the embeddings tends to a uniform distribution, in line with the goal of RankMe.

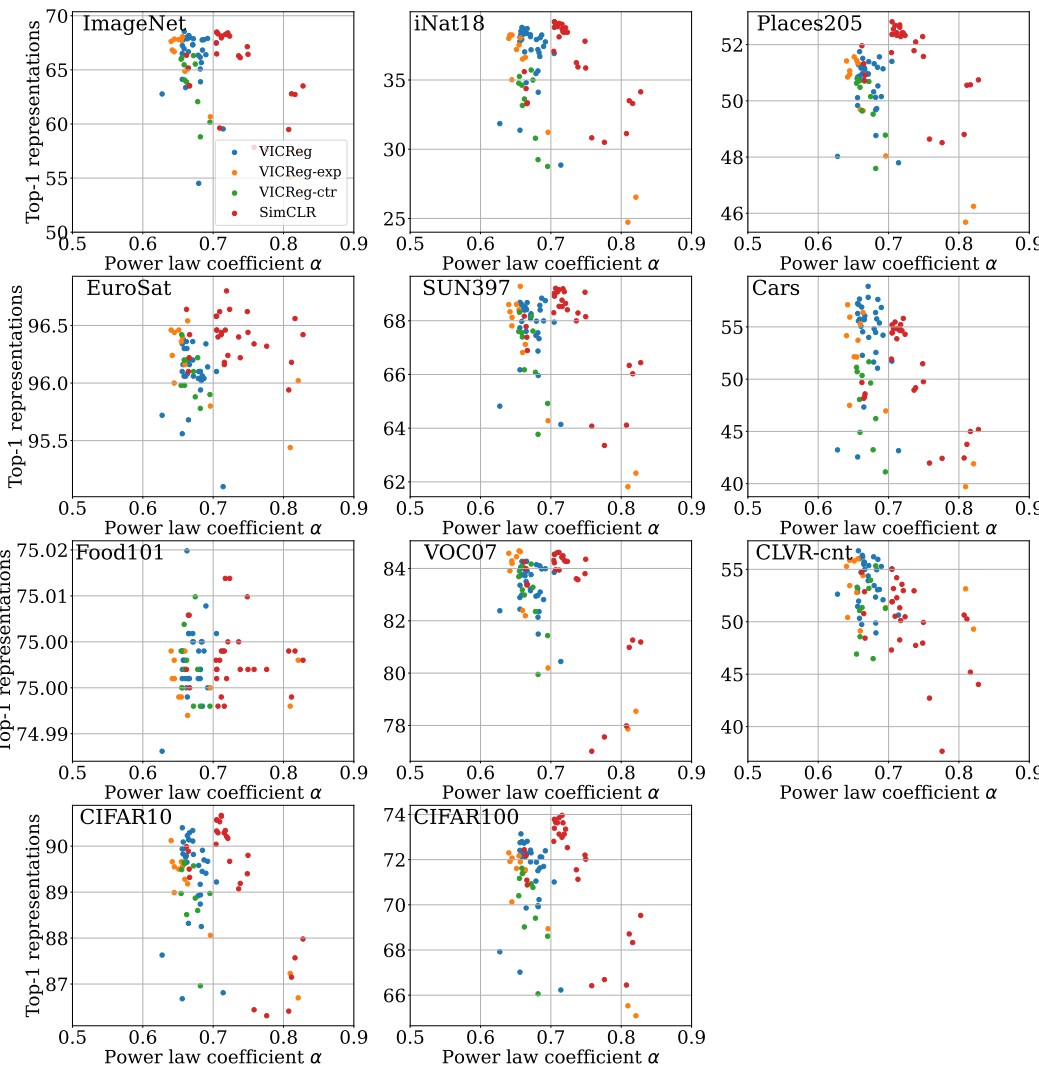

Figure S6: Link between $\alpha$-ReQ measured on the representations and performance on the representations.

As we can see in figs. S8 and S9, the power-law prior of $\alpha$-ReQ holds well in the case of non-collapsed embeddings, but when we apply it on collapsed ones, this assumptions fails. It even provides a poor approximation of the main rank "plateau" with the highest singular values as can be seen on the right

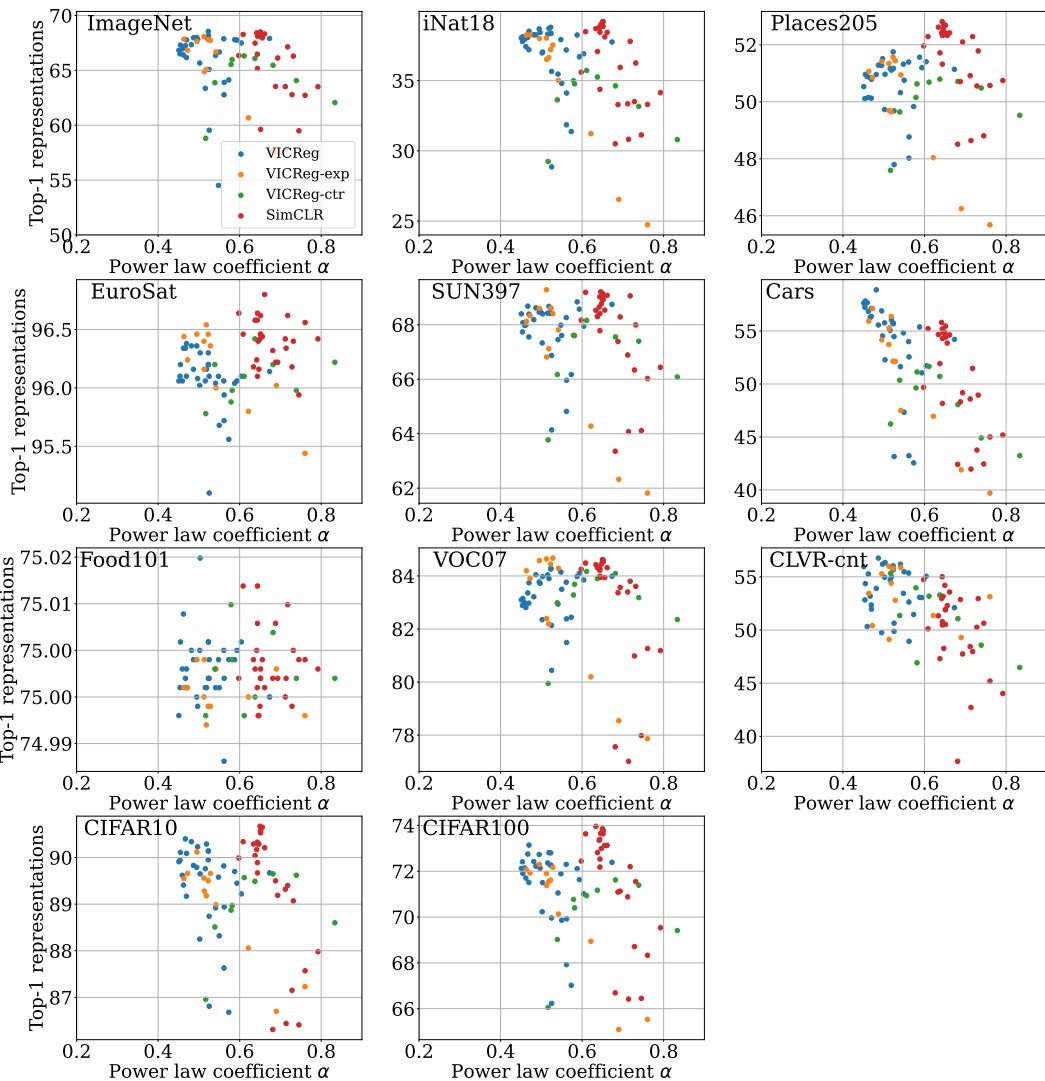

Figure S7: Link between $\alpha$-ReQ measured on the embeddings and performance on the representations.

of fig. S9. This further confirms the findings of He & Ozay (2022), and shows that when applying $\alpha$-ReQ directly on the embeddings one must be careful since the core assumptions of the method is violated.

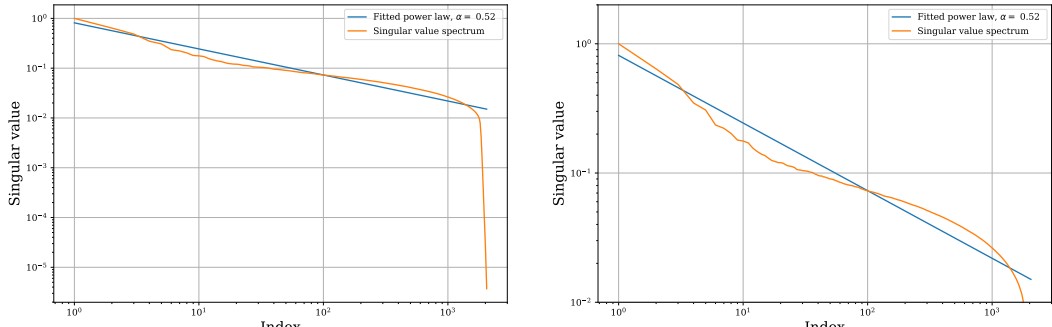

Figure S8: Validation of the power-law prior on un-collapsed representations.(Left) Overall visualization. (Right) Zoom on the high singular values.

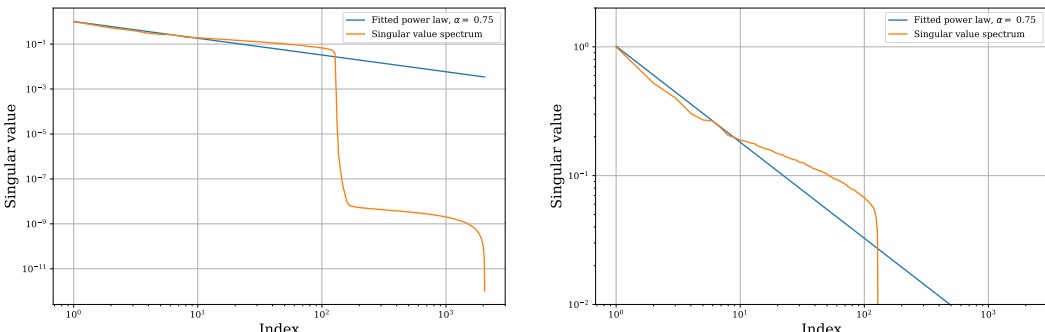

Figure S9: The power-law prior does not hold on collapsed representations.(Left) Overall visualization. (Right) Zoom on the high singular values.

## F    COMPARISON OF THE RANK ESTIMATORS

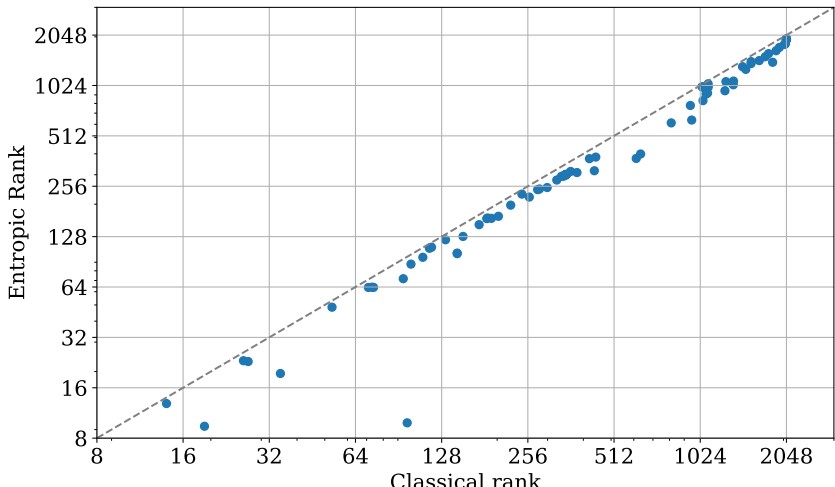

Figure S10: Relationship between the two rank estimators, Pearson correlation coefficient of $0.99$. Outliers indicate embeddings with singular values to the threshold, showing how the entropic rank takes into account this information.

Since we do not rely on the classical threshold-based rank estimator, it is important to verify how well our entropy based one correlates with it. As we can see in fig. S10, both estimates discussed previously correlate extremely well, showing that using one or the other should not lead to significant differences, as validated in appendix H. Nonetheless, the entropic estimator takes into account the degree of whitening of the embeddings, which links better to theoretical results.

## G    CONVERGENCE OF THE RANK ESTIMATORS

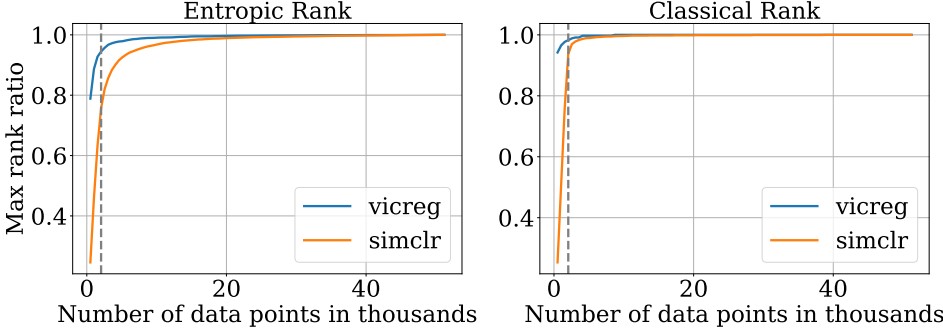

Figure S11: Convergence of the rank estimators on ImageNet as a function of the number of samples for 2048 dimensional outputs, as indicated by the vertical line.

As we can see in  fig. S11, the rank estimates converge extremely quickly, especially for VICReg. For both VICReg and SimCLR, 10000 samples are enough to obtain more than $95\%$ of the final rank. It is worth noting that the entropic rank estimator converges more slowly than the classical rank estimator, as it is more sensitive to the singular values. The fact that the rank can be approximated with few samples is encouraging for its use during training and not only as a measure of performance after pretraining.

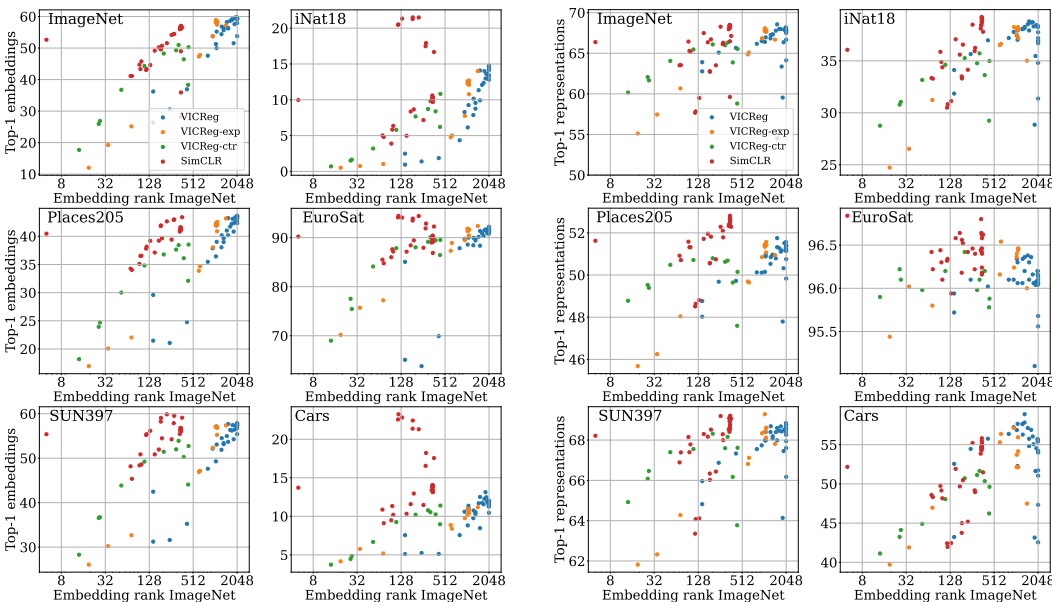

Figure S12: Reproduction of the top of fig. 2 with the classical rank estimator. Embeddings' rank transfers from source to target datasets. The estimates used 25600 images from the respective datasets.

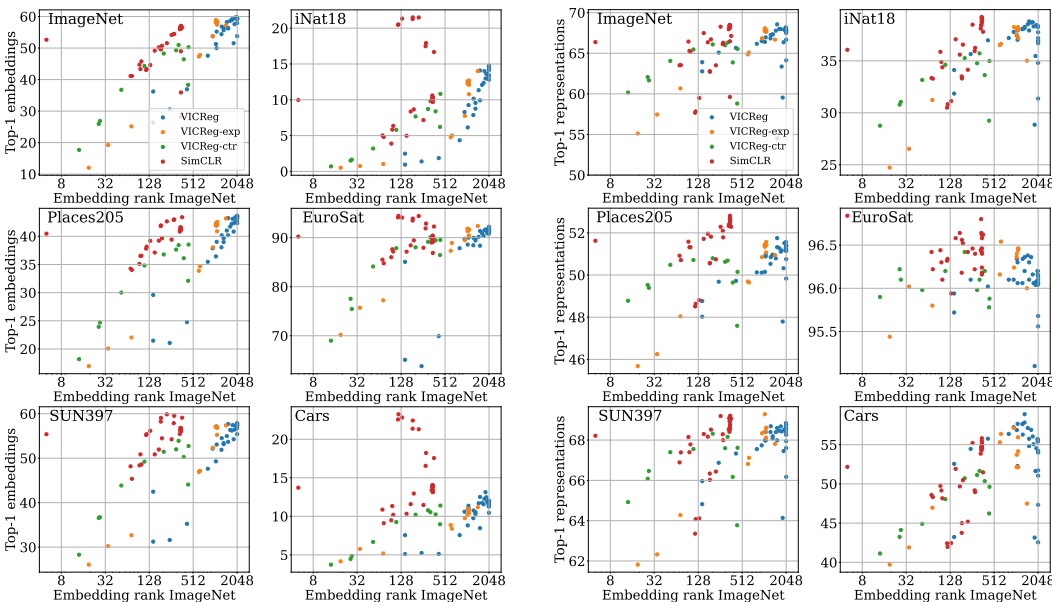

Figure S13: Reproduction of fig. 3 with the classical rank estimator.**(Left)** Validation of RankMe on embeddings, a higher ImageNet rank leads to improved performance across methods and datasets.**(Right)** Validation of RankMe on representations, where the link is even clearer, reinforcing RankMe's practical use.

## H    REPRODUCTION OF FIGURES WITH THE CLASSICAL RANK ESTIMATOR

As can be seen in figs. S12 and S13, the results that we obtain using the classical threshold-based rank estimator are extremely similar to the ones with the entropic estimator. The exact values do differ, but the behaviors stay the same. One of the main differences is illustrated in fig. S13, where we can see that the target rank is almost identical to the source one when we previously saw a drop of around $50\%$. This can be explained by the fact that some features may be less present in the target dataset, reducing the associated singular values, and thus the entropic rank.

All of this shows that using one or the other will lead to similar results in practical scenarios.

## I    DETAILED TRAINING AND EVALUATION PROCEDURES

### I.1    PRETRAINING

All pretrainings were done with ResNet-50 backbones. The projector used is a MLP with intermediate dimensions $8192, 8192, 2048$. They were trained with the LARS optimizer using a momentum of $0.9$, weight decay $10^{-6}$ and varying learning rates depending on the method. VICReg used $0.3$ base learning rate, SimCLR $0.5$ or $0.6$ depending on the experiment, VICReg-exp $0.6$ and VICReg-ctr $0.6$.

Table S1: Image augmentation parameters, taken from Grill et al. (2020).

| Parameter | View 1 | View 2 |
|---|---|---|
| Random crop probability | 1.0 | 1.0 |
| Horizontal flip probability | 0.5 | 0.5 |
| Color jittering probability | 0.8 | 0.8 |
| Brightness adjustment max intensity | 0.4 | 0.4 |
| Contrast adjustment max intensity | 0.4 | 0.4 |
| Saturation adjustment max intensity | 0.2 | 0.2 |
| Hue adjustment max intensity | 0.1 | 0.1 |
| Grayscale probability | 0.2 | 0.2 |
| Gaussian blurring probability | 1.0 | 0.1 |
| Solarization probability. | 0.0 | 0.2 |

The learning rate is then computed as $lr = base\_lr * batch\_size/256$. We do a 10-epochs linear warmup and then use cosine annealing. We used batch sizes of 2048 for SimCLR and 1024 for other methods. SimCLR and VICReg-ctr also use a default temperature of 0.15, and 0.1 for VICReg-exp. We used the image augmentation strategy from Grill et al. (2020) illustrated in table S1.

## I.2 EVALUATION

Table S2: Optimization parameters used to evaluate on downstream datasets

| Dataset | Optimizer | Weight decay | Momentum | Learning rate | Epochs |
|---|---|---|---|---|---|
| iNaturalist18 | SGD (w/ Nesterov) | 0.0005 | 0.9 | 0.01 | 84 |
| Places205 | SGD (w/ Nesterov) | 0.0005 | 0.9 | 0.01 | 14 |
| EuroSat | SGD (w/ Nesterov) | 0.0005 | 0.9 | 0.01 | 28 |
| Sun397 | SGD (w/ Nesterov) | 0.0005 | 0.9 | 0.01 | 28 |
| StanfordCars | SGD (w/ Nesterov) | 0.0005 | 0.9 | 0.1 | 28 |
| CIFAR10 | SGD (w/ Nesterov) | 0.0005 | 0.9 | 0.1 | 28 |
| CIFAR100 | SGD (w/ Nesterov) | 0.0005 | 0.9 | 0.1 | 28 |
| CLEVR-count | SGD (w/ Nesterov) | 0.0005 | 0.9 | 0.1 | 50 |
| Food101 | SGD (w/ Nesterov) | 0.0005 | 0.9 | 0.1 | 28 |
| VOC07 | N/A, see in text | | | | |

For all datasets except StanfordCars, we use the standard protocol in VISSL. On StanfordCars we mostly tuned the learning rate. The parameters that we used are described in table S2. For data augmentation, we used random resized crops and random horizontal flips during training, and center crop for evaluation. For VOC07, we follow the common protocol using SVMs, as used in Bardes et al. (2021). We use the default VISSL settings for this evaluation.

## J   DETAILED TABLES FOR HYPERPARAMETER SELECTION

Table S3: Top-1 accuracies computed on representations when tuning hyperparameters with ImageNet validation performance or with RankMe.

| Dataset | Method | VICReg | | | | SimCLR | | |
|---|---|---|---|---|---|---|---|---|
| | | cov. | inv. | LR | WD | temp. | LR. | WD. |
| ImageNet | ImageNet Oracle | 68.2 | 68.2 | 68.6 | 68.0 | 68.5 | 68.5 | 68.3 |
| | RankMe | 67.8 | 67.9 | 68.2 | 67.8 | 67.1 | 68.0 | 68.3 |
| | $\alpha$-ReQ | 67.9 | 67.5 | 59.5 | 67.8 | 63.5 | 68.1 | 32.3 |
| iNat18 | ImageNet Oracle | 38.4 | 38.4 | 38.8 | 38.3 | 39.2 | 39.2 | 38.9 |
| | RankMe | 36.7 | 37.2 | 38.4 | 38.3 | 37.8 | 38.1 | 38.9 |
| | $\alpha$-ReQ | 37.8 | 36.9 | 28.9 | 38.3 | 34.1 | 38.4 | 38.7 |
| Places205 | ImageNet Oracle | 51.2 | 51.2 | 51.8 | 51.3 | 52.4 | 52.4 | 52.6 |
| | RankMe | 51.2 | 51.4 | 51.2 | 51.6 | 52.3 | 52.3 | 52.6 |
| | $\alpha$-ReQ | 51.1 | 51.4 | 47.8 | 51.6 | 50.7 | 52.3 | 52.6 |
| EuroSat | ImageNet Oracle | 96.2 | 96.2 | 96.3 | 96.2 | 96.5 | 96.5 | 96.4 |
| | RankMe | 96.1 | 96.1 | 96.2 | 96.0 | 96.6 | 96.4 | 96.4 |
| | $\alpha$-ReQ | 96.1 | 96.1 | 95.1 | 96.0 | 96.4 | 96.6 | 96.2 |
| SUN397 | ImageNet Oracle | 68.4 | 68.4 | 68.6 | 68.6 | 68.9 | 68.9 | 69.2 |
| | RankMe | 68.6 | 68.3 | 68.4 | 68.8 | 69.1 | 68.5 | 69.2 |
| | $\alpha$-ReQ | 68.7 | 67.9 | 64.1 | 68.8 | 66.4 | 68.4 | 68.5 |
| Cars | ImageNet Oracle | 55.7 | 55.7 | 55.8 | 55.6 | 54.4 | 54.4 | 54.9 |
| | RankMe | 51.1 | 54.0 | 55.7 | 55.4 | 51.5 | 53.9 | 54.9 |
| | $\alpha$-ReQ | 54.2 | 51.7 | 43.2 | 55.4 | 45.2 | 54.3 | 54.7 |
| FOOD101 | ImageNet Oracle | 75.0 | 75.0 | 75.0 | 75.0 | 75.0 | 75.0 | 75.0 |
| | RankMe | 75.0 | 75.0 | 75.0 | 75.0 | 75.0 | 75.0 | 75.0 |
| | $\alpha$-ReQ | 75.0 | 75.0 | 75.0 | 75.0 | 75.0 | 75.0 | 75.0 |
| VOC07 | ImageNet Oracle | 84.3 | 84.3 | 84.3 | 84.0 | 84.5 | 84.5 | 83.9 |
| | RankMe | 84.1 | 83.8 | 84.3 | 84.0 | 83.8 | 83.9 | 83.9 |
| | $\alpha$-ReQ | 84.0 | 83.9 | 80.4 | 84.0 | 81.2 | 84.3 | 84.4 |
| CLEVR-Count | ImageNet Oracle | 55.7 | 55.7 | 56.0 | 56.8 | 51.9 | 51.9 | 53.2 |
| | RankMe | 53.0 | 55.4 | 55.7 | 53.1 | 48.0 | 52.3 | 53.2 |
| | $\alpha$-ReQ | 52.1 | 55.1 | 50.6 | 53.1 | 44.0 | 50.5 | 51.3 |
| CIFAR10 | ImageNet Oracle | 90.1 | 90.1 | 90.0 | 89.8 | 90.6 | 90.6 | 90.3 |
| | RankMe | 89.5 | 89.8 | 90.1 | 89.7 | 89.4 | 90.6 | 90.3 |
| | $\alpha$-ReQ | 89.7 | 89.2 | 86.8 | 89.7 | 88.0 | 89.7 | 90.3 |
| CIFAR100 | ImageNet Oracle | 72.3 | 72.3 | 72.8 | 72.2 | 73.8 | 73.8 | 73.7 |
| | RankMe | 71.6 | 72.3 | 72.3 | 72.1 | 72.2 | 73.1 | 73.7 |
| | $\alpha$-ReQ | 72.4 | 71.0 | 66.2 | 72.1 | 69.5 | 72.5 | 74.0 |
| Average | ImageNet Oracle | 68.7 | 68.7 | 68.9 | 68.7 | 68.7 | 68.7 | 68.8 |
| | RankMe | 67.7 | 68.3 | 68.7 | 68.3 | 67.5 | 68.4 | 68.8 |
| | $\alpha$-ReQ | 68.1 | 67.8 | 63.4 | 68.3 | 64.9 | 68.2 | 65.3 |

Table S4: Top-1 accuracies computed on embeddings when tuning hyperparameters with ImageNet validation performance or with RankMe.

| Dataset | Method | VICReg | | | | SimCLR | | |
|---|---|---|---|---|---|---|---|---|
| | | cov. | inv. | LR | WD | temp. | LR. | WD. |
| ImageNet | ImageNet Oracle | 59.7 | 59.7 | 59.7 | 59.7 | 56.9 | 56.9 | 57.1 |
| | RankMe | 59.6 | 59.7 | 59.7 | 59.5 | 56.5 | 56.0 | 57.1 |
| | $\alpha$-ReQ | 59.6 | 59.2 | 36.2 | 59.3 | 51.5 | 56.4 | 49.0 |
| iNat18 | ImageNet Oracle | 13.5 | 14.2 | 13.5 | 13.6 | 10.3 | 10.3 | 10.1 |
| | RankMe | 14.2 | 14.2 | 13.5 | 13.4 | 16.7 | 9.9 | 10.1 |
| | $\alpha$-ReQ | 14.2 | 14.8 | 2.5 | 13.2 | 21.5 | 10.0 | 10.0 |
| Places205 | ImageNet Oracle | 42.7 | 43.3 | 42.7 | 43.4 | 41.2 | 41.2 | 41.2 |
| | RankMe | 43.2 | 43.3 | 42.7 | 42.7 | 43.4 | 40.8 | 41.2 |
| | $\alpha$-ReQ | 43.2 | 43.6 | 29.6 | 42.9 | 42.6 | 41.0 | 41.5 |
| EuroSat | ImageNet Oracle | 91.3 | 91.7 | 91.3 | 91.0 | 90.4 | 90.4 | 89.5 |
| | RankMe | 91.0 | 91.7 | 91.3 | 91.3 | 92.3 | 89.0 | 89.5 |
| | $\alpha$-ReQ | 91.0 | 91.4 | 85.1 | 90.8 | 94.4 | 89.6 | 89.8 |
| SUN397 | ImageNet Oracle | 57.3 | 57.0 | 57.3 | 57.3 | 56.4 | 56.4 | 56.2 |
| | RankMe | 57.4 | 57.0 | 57.3 | 56.7 | 59.1 | 55.4 | 56.2 |
| | $\alpha$-ReQ | 57.4 | 57.4 | 42.5 | 57.2 | 59.9 | 56.2 | 56.2 |
| Cars | ImageNet Oracle | 12.0 | 12.0 | 12.0 | 11.9 | 14.0 | 14.0 | 13.2 |
| | RankMe | 11.6 | 12.0 | 12.0 | 11.5 | 17.6 | 13.4 | 13.2 |
| | $\alpha$-ReQ | 11.6 | 12.0 | 7.5 | 11.3 | 21.3 | 13.9 | 13.5 |
| FOOD101 | ImageNet Oracle | 75.0 | 75.0 | 75.0 | 75.0 | 75.0 | 75.0 | 75.0 |
| | RankMe | 75.0 | 75.0 | 75.0 | 75.0 | 75.0 | 75.0 | 75.0 |
| | $\alpha$-ReQ | 75.0 | 75.0 | 75.0 | 75.0 | 75.0 | 75.0 | 75.0 |
| VOC07 | ImageNet Oracle | 79.5 | 79.2 | 79.5 | 79.7 | 79.7 | 79.7 | 79.7 |
| | RankMe | 79.2 | 79.2 | 79.5 | 79.3 | 78.5 | 79.3 | 79.7 |
| | $\alpha$-ReQ | 79.2 | 79.2 | 73.1 | 79.6 | 76.8 | 79.5 | 79.9 |
| CLEVR-Count | ImageNet Oracle | 43.9 | 44.4 | 43.9 | 46.1 | 43.5 | 43.5 | 46.0 |
| | RankMe | 43.9 | 44.4 | 43.9 | 43.0 | 43.0 | 44.8 | 46.0 |
| | $\alpha$-ReQ | 43.9 | 43.8 | 41.7 | 44.9 | 37.0 | 45.2 | 45.9 |
| CIFAR10 | ImageNet Oracle | 80.4 | 81.2 | 80.4 | 79.7 | 79.3 | 79.3 | 79.8 |
| | RankMe | 80.6 | 81.2 | 80.4 | 80.3 | 79.5 | 79.5 | 79.8 |
| | $\alpha$-ReQ | 80.6 | 81.0 | 72.5 | 79.6 | 79.2 | 78.5 | 79.4 |
| CIFAR100 | ImageNet Oracle | 52.8 | 53.3 | 52.8 | 52.9 | 52.6 | 52.6 | 52.2 |
| | RankMe | 53.8 | 53.3 | 52.8 | 52.5 | 54.0 | 52.2 | 52.2 |
| | $\alpha$-ReQ | 53.8 | 53.9 | 41.5 | 52.2 | 56.5 | 52.0 | 52.3 |
| Average | ImageNet Oracle | 55.3 | 55.5 | 55.3 | 55.5 | 54.5 | 54.5 | 54.5 |
| | RankMe | 55.4 | 55.5 | 55.3 | 55.0 | 56.0 | 54.1 | 54.5 |
| | $\alpha$-ReQ | 55.4 | 55.6 | 46.1 | 55.1 | 56.0 | 54.3 | 53.9 |

## K  COMPLETE PERFORMANCE TABLES

Table S5: Hyperparameters for all runs.

| Method | Run | Batch size | Learning rate | Weight decay | Loss hyperparameters |
|---|---|---|---|---|---|
| VICReg | 0 | 1024 | 0.3 | $10^{-6}$ | $\lambda : 25, \mu : 25, \nu : 0.3$ |
| | 1 | 1024 | 0.3 | $10^{-6}$ | $\lambda : 25, \mu : 25, \nu : 0.4$ |
| | 2 | 1024 | 0.3 | $10^{-6}$ | $\lambda : 25, \mu : 25, \nu : 0.5$ |
| | 3 | 1024 | 0.3 | $10^{-6}$ | $\lambda : 25, \mu : 25, \nu : 0.6$ |
| | 4 | 1024 | 0.3 | $10^{-6}$ | $\lambda : 25, \mu : 25, \nu : 0.7$ |
| | 5 | 1024 | 0.3 | $10^{-6}$ | $\lambda : 25, \mu : 25, \nu : 0.8$ |
| | 6 | 1024 | 0.3 | $10^{-6}$ | $\lambda : 25, \mu : 25, \nu : 0.9$ |
| | 7 | 1024 | 0.3 | $10^{-6}$ | $\lambda : 25, \mu : 25, \nu : 1$ |
| | 8 | 1024 | 0.3 | $10^{-6}$ | $\lambda : 25, \mu : 25, \nu : 2$ |
| | 9 | 1024 | 0.3 | $10^{-6}$ | $\lambda : 25, \mu : 25, \nu : 4$ |
| | 10 | 1024 | 0.3 | $10^{-6}$ | $\lambda : 25, \mu : 25, \nu : 8$ |
| | 11 | 1024 | 0.3 | $10^{-6}$ | $\lambda : 25, \mu : 25, \nu : 16$ |
| | 12 | 1024 | 0.3 | $10^{-6}$ | $\lambda : 5, \mu : 25, \nu : 4$ |
| | 13 | 1024 | 0.3 | $10^{-6}$ | $\lambda : 10, \mu : 25, \nu : 4$ |
| | 14 | 1024 | 0.3 | $10^{-6}$ | $\lambda : 15, \mu : 25, \nu : 4$ |
| | 15 | 1024 | 0.3 | $10^{-6}$ | $\lambda : 20, \mu : 25, \nu : 4$ |
| | 16 | 1024 | 0.3 | $10^{-6}$ | $\lambda : 30, \mu : 25, \nu : 4$ |
| | 17 | 1024 | 0.3 | $10^{-6}$ | $\lambda : 35, \mu : 25, \nu : 4$ |
| | 18 | 1024 | 0.3 | $10^{-6}$ | $\lambda : 40, \mu : 25, \nu : 4$ |
| | 19 | 1024 | 0.3 | $10^{-6}$ | $\lambda : 45, \mu : 25, \nu : 4$ |
| | 20 | 1024 | 0.3 | $10^{-6}$ | $\lambda : 50, \mu : 25, \nu : 4$ |
| | 21 | 1024 | 0.1 | $10^{-6}$ | $\lambda : 25, \mu : 25, \nu : 4$ |
| | 22 | 1024 | 0.2 | $10^{-6}$ | $\lambda : 25, \mu : 25, \nu : 4$ |
| | 23 | 1024 | 0.3 | $10^{-6}$ | $\lambda : 25, \mu : 25, \nu : 4$ |
| | 24 | 1024 | 0.4 | $10^{-6}$ | $\lambda : 25, \mu : 25, \nu : 4$ |
| | 25 | 1024 | 0.5 | $10^{-6}$ | $\lambda : 25, \mu : 25, \nu : 4$ |
| | 26 | 1024 | 0.3 | $10^{-7}$ | $\lambda : 25, \mu : 25, \nu : 4$ |
| | 27 | 1024 | 0.3 | $10^{-6}$ | $\lambda : 25, \mu : 25, \nu : 4$ |
| | 28 | 1024 | 0.3 | $10^{-5}$ | $\lambda : 25, \mu : 25, \nu : 4$ |
| | 29 | 1024 | 0.3 | $10^{-4}$ | $\lambda : 25, \mu : 25, \nu : 4$ |
| | 30 | 1024 | 0.3 | $10^{-3}$ | $\lambda : 25, \mu : 25, \nu : 4$ |
| | 31 | 1024 | 0.3 | $10^{-2}$ | $\lambda : 25, \mu : 25, \nu : 4$ |
| VICReg-exp | 0 | 1024 | 0.5 | $10^{-6}$ | $\lambda : 1, \mu : 1, \nu : 2, \tau : 0.05$ |
| | 1 | 1024 | 0.5 | $10^{-6}$ | $\lambda : 1, \mu : 1, \nu : 2, \tau : 0.07$ |
| | 2 | 1024 | 0.5 | $10^{-6}$ | $\lambda : 1, \mu : 1, \nu : 2, \tau : 0.1$ |
| | 3 | 1024 | 0.5 | $10^{-6}$ | $\lambda : 1, \mu : 1, \nu : 2, \tau : 0.2$ |
| | 4 | 1024 | 0.5 | $10^{-6}$ | $\lambda : 1, \mu : 1, \nu : 2, \tau : 0.3$ |
| | 5 | 1024 | 0.5 | $10^{-6}$ | $\lambda : 1, \mu : 1, \nu : 2, \tau : 0.4$ |
| | 6 | 1024 | 0.5 | $10^{-6}$ | $\lambda : 1, \mu : 1, \nu : 0.1, \tau : 0.1$ |
| | 7 | 1024 | 0.5 | $10^{-6}$ | $\lambda : 1, \mu : 1, \nu : 0.5, \tau : 0.1$ |
| | 8 | 1024 | 0.5 | $10^{-6}$ | $\lambda : 1, \mu : 1, \nu : 1, \tau : 0.1$ |
| | 9 | 1024 | 0.5 | $10^{-6}$ | $\lambda : 1, \mu : 1, \nu : 4, \tau : 0.1$ |
| | 10 | 1024 | 0.5 | $10^{-6}$ | $\lambda : 1, \mu : 1, \nu : 8, \tau : 0.1$ |
| | 11 | 1024 | 0.5 | $10^{-6}$ | $\lambda : 1, \mu : 1, \nu : 16, \tau : 0.1$ |

Table S6: Hyperparameters for all runs, continued.

| Method | Run | Batch size | Learning rate | Weight decay | Loss hyperparameters |
|---|---|---|---|---|---|
| | 0 | 1024 | 0.5 | $10^{-6}$ | $\lambda : 1, \mu : 1, \nu : 1, \tau : 0.05$ |
| | 1 | 1024 | 0.5 | $10^{-6}$ | $\lambda : 1, \mu : 1, \nu : 1, \tau : 0.07$ |
| | 2 | 1024 | 0.5 | $10^{-6}$ | $\lambda : 1, \mu : 1, \nu : 1, \tau : 0.1$ |
| | 3 | 1024 | 0.5 | $10^{-6}$ | $\lambda : 1, \mu : 1, \nu : 1, \tau : 0.2$ |
| | 4 | 1024 | 0.5 | $10^{-6}$ | $\lambda : 1, \mu : 1, \nu : 1, \tau : 0.3$ |
| VICReg-ctr | 5 | 1024 | 0.5 | $10^{-6}$ | $\lambda : 1, \mu : 1, \nu : 1, \tau : 0.4$ |
| | 6 | 1024 | 0.5 | $10^{-6}$ | $\lambda : 1, \mu : 1, \nu : 0.1, \tau : 0.1$ |
| | 7 | 1024 | 0.5 | $10^{-6}$ | $\lambda : 1, \mu : 1, \nu : 0.5, \tau : 0.1$ |
| | 8 | 1024 | 0.5 | $10^{-6}$ | $\lambda : 1, \mu : 1, \nu : 2, \tau : 0.1$ |
| | 9 | 1024 | 0.5 | $10^{-6}$ | $\lambda : 1, \mu : 1, \nu : 4, \tau : 0.1$ |
| | 10 | 1024 | 0.5 | $10^{-6}$ | $\lambda : 1, \mu : 1, \nu : 8, \tau : 0.1$ |
| | 0 | 2048 | 0.6 | $10^{-6}$ | $d : 512, \tau : 0.05$ |
| | 1 | 2048 | 0.6 | $10^{-6}$ | $d : 512, \tau : 0.07$ |
| | 2 | 2048 | 0.6 | $10^{-6}$ | $d : 512, \tau : 0.1$ |
| | 3 | 2048 | 0.6 | $10^{-6}$ | $d : 512, \tau : 0.2$ |
| | 4 | 2048 | 0.6 | $10^{-6}$ | $d : 512, \tau : 0.3$ |
| | 5 | 2048 | 0.6 | $10^{-6}$ | $d : 512, \tau : 0.4$ |
| | 6 | 2048 | 0.6 | $10^{-6}$ | $d : 2048, \tau : 0.05$ |
| | 7 | 2048 | 0.6 | $10^{-6}$ | $d : 2048, \tau : 0.07$ |
| | 8 | 2048 | 0.6 | $10^{-6}$ | $d : 2048, \tau : 0.1$ |
| | 9 | 2048 | 0.6 | $10^{-6}$ | $d : 2048, \tau : 0.2$ |
| | 10 | 2048 | 0.6 | $10^{-6}$ | $d : 2048, \tau : 0.3$ |
| | 11 | 2048 | 0.6 | $10^{-6}$ | $d : 2048, \tau : 0.4$ |
| | 12 | 2048 | 0.5 | $10^{-6}$ | $d : 2048, \tau : 0.05$ |
| | 13 | 2048 | 0.5 | $10^{-6}$ | $d : 2048, \tau : 0.07$ |
| SimCLR | 14 | 2048 | 0.5 | $10^{-6}$ | $d : 2048, \tau : 0.1$ |
| | 15 | 2048 | 0.5 | $10^{-6}$ | $d : 2048, \tau : 0.15$ |
| | 16 | 2048 | 0.5 | $10^{-6}$ | $d : 2048, \tau : 0.2$ |
| | 17 | 2048 | 0.5 | $10^{-6}$ | $d : 2048, \tau : 0.3$ |
| | 18 | 2048 | 0.5 | $10^{-6}$ | $d : 2048, \tau : 0.4$ |
| | 19 | 2048 | 0.5 | $10^{-7}$ | $d : 2048, \tau : 0.15$ |
| | 20 | 2048 | 0.5 | $10^{-6}$ | $d : 2048, \tau : 0.15$ |
| | 21 | 2048 | 0.5 | $10^{-5}$ | $d : 2048, \tau : 0.15$ |
| | 22 | 2048 | 0.5 | $10^{-4}$ | $d : 2048, \tau : 0.15$ |
| | 23 | 2048 | 0.5 | $10^{-3}$ | $d : 2048, \tau : 0.15$ |
| | 24 | 2048 | 0.5 | $10^{-2}$ | $d : 2048, \tau : 0.15$ |
| | 25 | 2048 | 0.2 | $10^{-6}$ | $d : 2048, \tau : 0.15$ |
| | 26 | 2048 | 0.3 | $10^{-6}$ | $d : 2048, \tau : 0.15$ |
| | 27 | 2048 | 0.4 | $10^{-6}$ | $d : 2048, \tau : 0.15$ |
| | 28 | 2048 | 0.5 | $10^{-6}$ | $d : 2048, \tau : 0.15$ |
| | 29 | 2048 | 0.6 | $10^{-6}$ | $d : 2048, \tau : 0.15$ |
| | 30 | 2048 | 0.8 | $10^{-6}$ | $d : 2048, \tau : 0.15$ |

Table S7: Top-1 on representations in all settings.

| Method | Run | ImageNet | iNat18 | Places205 | EuroSat | SUN397 | Cars |
|--------|-----|----------|--------|-----------|---------|--------|------|
| VICReg | 0 | 63.90 | 34.12 | 48.77 | 95.94 | 65.96 | 52.56 |
| | 1 | 65.08 | 35.65 | 49.68 | 96.10 | 66.87 | 54.47 |
| | 2 | 65.67 | 36.97 | 49.73 | 96.02 | 67.33 | 55.76 |
| | 3 | 66.17 | 37.20 | 50.13 | 96.10 | 67.55 | 56.37 |
| | 4 | 66.40 | 37.42 | 50.15 | 96.34 | 67.99 | 56.86 |
| | 5 | 66.83 | 38.05 | 50.53 | 96.06 | 68.40 | 57.63 |
| | 6 | 67.30 | 38.13 | 50.96 | 96.20 | 68.08 | 57.83 |
| | 7 | 67.34 | 38.26 | 50.96 | 96.36 | 68.19 | 58.89 |
| | 8 | 68.00 | 38.68 | 51.28 | 96.36 | 68.46 | 56.90 |
| | 9 | 68.16 | 38.36 | 51.17 | 96.20 | 68.42 | 55.70 |
| | 10 | 67.91 | 37.75 | 51.14 | 96.14 | 68.75 | 54.21 |
| | 11 | 67.77 | 36.70 | 51.20 | 96.06 | 68.57 | 51.05 |
| | 12 | 64.12 | 31.37 | 49.83 | 95.56 | 66.17 | 42.56 |
| | 13 | 66.67 | 34.81 | 50.76 | 95.68 | 67.61 | 47.33 |
| | 14 | 67.49 | 36.91 | 51.40 | 96.10 | 67.95 | 51.72 |
| | 15 | 67.87 | 37.18 | 51.40 | 96.06 | 68.26 | 54.00 |
| | 16 | 67.99 | 38.71 | 51.11 | 96.16 | 68.68 | 56.05 |
| | 17 | 67.78 | 38.52 | 50.79 | 96.38 | 68.39 | 57.13 |
| | 18 | 67.25 | 38.08 | 50.85 | 96.34 | 68.69 | 56.29 |
| | 19 | 66.95 | 37.93 | 50.88 | 96.06 | 67.98 | 57.67 |
| | 20 | 66.51 | 37.79 | 50.11 | 96.10 | 67.74 | 57.23 |
| | 21 | 59.54 | 28.85 | 47.80 | 95.10 | 64.14 | 43.15 |
| | 22 | 66.36 | 35.47 | 50.32 | 96.04 | 67.45 | 51.64 |
| | 23 | 68.16 | 38.36 | 51.17 | 96.20 | 68.42 | 55.70 |
| | 24 | 68.56 | 38.80 | 51.75 | 96.30 | 68.60 | 55.75 |
| | 25 | 62.77 | 31.85 | 48.02 | 95.72 | 64.82 | 43.23 |
| | 26 | 67.79 | 38.25 | 51.57 | 96.04 | 68.84 | 55.38 |
| | 27 | 67.97 | 38.26 | 51.29 | 96.16 | 68.62 | 55.57 |
| | 28 | 67.87 | 38.43 | 51.51 | 96.08 | 68.52 | 54.53 |
| | 29 | 63.36 | 38.31 | 51.17 | 96.06 | 68.43 | 55.06 |
| | 30 | 54.52 | 37.92 | 51.32 | 96.10 | 67.99 | 54.82 |
| | 31 | 40.73 | 37.03 | 50.97 | 96.30 | 68.40 | 52.28 |
| VICReg-exp | 0 | 67.74 | 37.53 | 51.44 | 96.36 | 68.41 | 52.12 |
| | 1 | 67.64 | 38.00 | 51.42 | 96.46 | 68.60 | 54.16 |
| | 2 | 67.84 | 38.25 | 51.07 | 96.44 | 68.12 | 55.94 |
| | 3 | 65.09 | 36.64 | 49.65 | 96.54 | 67.12 | 56.37 |
| | 4 | 60.67 | 31.22 | 48.04 | 95.80 | 64.28 | 46.96 |
| | 5 | 57.46 | 26.54 | 46.25 | 96.02 | 62.33 | 41.90 |
| | 6 | 55.12 | 24.73 | 45.68 | 95.44 | 61.82 | 39.71 |
| | 7 | 64.87 | 36.51 | 49.69 | 96.16 | 66.82 | 55.30 |
| | 8 | 66.84 | 38.25 | 50.85 | 96.24 | 68.34 | 57.12 |
| | 9 | 68.08 | 38.03 | 51.34 | 96.40 | 69.28 | 53.72 |
| | 10 | 67.80 | 37.20 | 51.57 | 96.46 | 68.61 | 52.15 |
| | 11 | 66.68 | 35.02 | 50.94 | 96.00 | 67.81 | 47.49 |

Table S8: Top-1 on representations in all settings, continued.

| Method | Run | ImageNet | iNat18 | Places205 | EuroSat | SUN397 | Cars |
|---|---|---|---|---|---|---|---|
| VICReg-ctr | 0 | 65.54 | 35.00 | 50.15 | 95.88 | 67.62 | 49.63 |
| | 1 | 66.32 | 35.72 | 50.69 | 96.10 | 68.16 | 51.66 |
| | 2 | 66.09 | 35.26 | 50.80 | 96.42 | 68.32 | 50.72 |
| | 3 | 64.06 | 33.16 | 50.48 | 95.98 | 67.40 | 44.91 |
| | 4 | 62.06 | 30.80 | 49.53 | 96.22 | 66.08 | 43.24 |
| | 5 | 60.17 | 28.76 | 48.78 | 95.90 | 64.92 | 41.13 |
| | 6 | 61.66 | 31.05 | 49.40 | 96.10 | 66.47 | 44.12 |
| | 7 | 65.47 | 34.63 | 50.71 | 96.20 | 67.55 | 48.05 |
| | 8 | 65.99 | 34.77 | 50.63 | 95.98 | 67.60 | 51.14 |
| | 9 | 63.87 | 33.63 | 49.64 | 96.20 | 66.17 | 50.35 |
| | 10 | 58.81 | 29.24 | 47.60 | 95.78 | 63.77 | 46.23 |
| SimCLR | 0 | 57.68 | 30.50 | 48.51 | 96.32 | 63.36 | 42.42 |
| | 1 | 62.79 | 33.50 | 50.56 | 96.18 | 66.34 | 43.76 |
| | 2 | 66.13 | 35.94 | 52.10 | 96.22 | 68.29 | 49.17 |
| | 3 | 66.35 | 35.60 | 51.96 | 96.64 | 68.17 | 49.68 |
| | 4 | 65.17 | 34.38 | 51.32 | 96.10 | 67.78 | 48.17 |
| | 5 | 63.54 | 33.29 | 50.71 | 96.22 | 67.39 | 48.31 |
| | 6 | 57.84 | 30.82 | 48.64 | 96.34 | 64.07 | 41.97 |
| | 7 | 62.73 | 33.30 | 50.57 | 96.56 | 66.03 | 44.99 |
| | 8 | 66.30 | 36.25 | 51.79 | 96.40 | 67.99 | 48.95 |
| | 9 | 66.71 | 36.56 | 51.82 | 96.52 | 68.52 | 50.47 |
| | 10 | 65.29 | 34.90 | 51.32 | 96.30 | 67.40 | 49.16 |
| | 11 | 63.52 | 33.35 | 50.92 | 96.42 | 66.89 | 48.59 |
| | 12 | 59.49 | 31.13 | 48.80 | 95.94 | 64.11 | 42.46 |
| | 13 | 63.51 | 34.14 | 50.75 | 96.42 | 66.44 | 45.18 |
| | 14 | 67.14 | 37.80 | 52.29 | 96.62 | 69.06 | 51.47 |
| | 15 | 68.48 | 39.20 | 52.37 | 96.46 | 68.92 | 54.43 |
| | 16 | 68.27 | 38.48 | 52.29 | 96.46 | 69.19 | 55.22 |
| | 17 | 67.48 | 37.07 | 51.72 | 96.58 | 68.30 | 51.92 |
| | 18 | 66.44 | 35.87 | 51.58 | 96.44 | 68.15 | 49.76 |
| | 19 | 68.33 | 38.93 | 52.56 | 96.40 | 69.21 | 54.86 |
| | 20 | 68.13 | 39.09 | 52.42 | 96.42 | 69.15 | 54.83 |
| | 21 | 66.47 | 38.80 | 52.81 | 96.58 | 69.03 | 55.19 |
| | 22 | 59.62 | 38.86 | 52.69 | 96.62 | 69.07 | 55.47 |
| | 23 | 47.58 | 39.03 | 52.70 | 96.16 | 68.77 | 54.96 |
| | 24 | 32.27 | 38.70 | 52.62 | 96.18 | 68.53 | 54.67 |
| | 25 | 66.37 | 36.06 | 51.62 | 96.84 | 68.22 | 52.17 |
| | 26 | 67.96 | 38.12 | 52.33 | 96.44 | 68.54 | 53.86 |
| | 27 | 68.32 | 38.44 | 52.42 | 96.80 | 69.08 | 54.63 |
| | 28 | 68.48 | 39.20 | 52.37 | 96.46 | 68.92 | 54.43 |
| | 29 | 68.41 | 38.77 | 52.42 | 96.24 | 68.65 | 55.81 |
| | 30 | 68.12 | 38.45 | 52.33 | 96.64 | 68.41 | 54.30 |

Table S9: Top-1 on representations in all settings, continued.

| Method | Run | ImageNet | CIFAR10 | CIFAR100 | FOOD101 | VOC07 | CLEVR-count |
|---|---|---|---|---|---|---|---|
| VICReg | 0 | 63.90 | 88.94 | 69.92 | 75.00 | 81.49 | 48.94 |
| | 1 | 65.08 | 88.74 | 69.96 | 75.00 | 82.14 | 49.88 |
| | 2 | 65.67 | 88.25 | 70.23 | 75.00 | 82.35 | 54.96 |
| | 3 | 66.17 | 89.17 | 71.51 | 75.00 | 82.97 | 52.35 |
| | 4 | 66.40 | 89.41 | 71.70 | 75.01 | 82.81 | 55.27 |
| | 5 | 66.83 | 89.91 | 72.12 | 75.00 | 83.10 | 55.95 |
| | 6 | 67.30 | 90.11 | 71.90 | 75.01 | 83.15 | 54.37 |
| | 7 | 67.34 | 90.34 | 72.42 | 75.00 | 83.21 | 53.92 |
| | 8 | 68.00 | 89.79 | 72.73 | 75.00 | 83.77 | 49.75 |
| | 9 | 68.16 | 90.14 | 72.26 | 75.01 | 84.27 | 55.69 |
| | 10 | 67.91 | 89.67 | 72.39 | 75.00 | 83.99 | 52.10 |
| | 11 | 67.77 | 89.45 | 71.63 | 75.00 | 84.10 | 53.05 |
| | 12 | 64.12 | 86.68 | 67.02 | 75.00 | 82.44 | 51.46 |
| | 13 | 66.67 | 88.32 | 69.86 | 75.00 | 83.50 | 55.48 |
| | 14 | 67.49 | 89.22 | 71.01 | 75.01 | 83.85 | 55.05 |
| | 15 | 67.87 | 89.82 | 72.30 | 75.00 | 83.76 | 55.36 |
| | 16 | 67.99 | 90.29 | 72.81 | 75.00 | 83.90 | 55.00 |
| | 17 | 67.78 | 90.09 | 73.14 | 75.00 | 83.74 | 51.97 |
| | 18 | 67.25 | 90.40 | 72.75 | 75.00 | 83.36 | 53.18 |
| | 19 | 66.95 | 89.62 | 72.14 | 75.00 | 82.99 | 50.33 |
| | 20 | 66.51 | 89.94 | 72.41 | 75.00 | 82.89 | 52.83 |
| | 21 | 59.54 | 86.81 | 66.23 | 75.00 | 80.44 | 50.64 |
| | 22 | 66.36 | 88.92 | 71.05 | 75.00 | 82.94 | 56.19 |
| | 23 | 68.16 | 90.14 | 72.26 | 75.01 | 84.27 | 55.69 |
| | 24 | 68.56 | 89.95 | 72.80 | 75.00 | 84.27 | 56.03 |
| | 25 | 62.77 | 87.63 | 67.92 | 74.99 | 82.38 | 52.63 |
| | 26 | 67.79 | 89.70 | 72.11 | 75.00 | 83.98 | 53.08 |
| | 27 | 67.97 | 89.83 | 72.22 | 75.00 | 84.05 | 56.77 |
| | 28 | 67.87 | 90.23 | 72.13 | 75.00 | 83.72 | 56.20 |
| | 29 | 63.36 | 89.76 | 72.36 | 75.00 | 84.04 | 54.71 |
| | 30 | 54.52 | 89.58 | 71.89 | 75.00 | 84.14 | 53.45 |
| | 31 | 40.73 | 89.65 | 71.50 | 75.01 | 83.97 | 56.34 |
| VICReg-exp | 0 | 67.74 | 89.66 | 72.17 | 75.00 | 84.67 | 52.79 |
| | 1 | 67.64 | 90.12 | 72.30 | 75.00 | 84.58 | 55.29 |
| | 2 | 67.84 | 89.55 | 72.07 | 75.00 | 84.20 | 53.45 |
| | 3 | 65.09 | 89.18 | 71.55 | 75.00 | 82.19 | 54.41 |
| | 4 | 60.67 | 88.06 | 68.94 | 75.00 | 80.20 | 51.35 |
| | 5 | 57.46 | 86.70 | 65.09 | 75.00 | 78.54 | 49.30 |
| | 6 | 55.12 | 87.23 | 65.53 | 75.00 | 77.87 | 53.14 |
| | 7 | 64.87 | 89.28 | 71.38 | 75.00 | 82.39 | 49.13 |
| | 8 | 66.84 | 89.66 | 71.92 | 75.00 | 83.91 | 50.41 |
| | 9 | 68.08 | 89.56 | 71.90 | 75.00 | 84.64 | 56.00 |
| | 10 | 67.80 | 89.50 | 71.61 | 75.00 | 84.45 | 55.80 |
| | 11 | 66.68 | 88.99 | 70.13 | 75.00 | 84.29 | 55.87 |

Table S10: Top-1 on representations in all settings, continued.

| Method | Run | ImageNet | CIFAR10 | CIFAR100 | FOOD101 | VOC07 | CLEVR-count |
|---|---|---|---|---|---|---|---|
| VICReg-ctr | 0 | 65.54 | 88.87 | 70.77 | 75.01 | 83.28 | 53.97 |
| | 1 | 66.32 | 89.57 | 70.93 | 75.00 | 84.17 | 53.19 |
| | 2 | 66.09 | 89.49 | 71.17 | 75.00 | 83.90 | 53.29 |
| | 3 | 64.06 | 89.62 | 71.39 | 75.00 | 83.18 | 48.57 |
| | 4 | 62.06 | 88.60 | 69.41 | 75.00 | 82.35 | 46.48 |
| | 5 | 60.17 | 88.97 | 68.61 | 75.00 | 81.43 | 51.27 |
| | 6 | 65.47 | 89.65 | 71.62 | 75.01 | 84.09 | 51.07 |
| | 7 | 65.99 | 88.97 | 70.40 | 75.00 | 83.69 | 46.92 |
| | 8 | 63.87 | 88.51 | 69.02 | 75.00 | 82.99 | 51.36 |
| | 9 | 58.81 | 86.96 | 66.06 | 75.00 | 79.95 | 55.33 |
| SimCLR | 0 | 57.68 | 86.31 | 66.69 | 75.00 | 77.56 | 37.65 |
| | 1 | 62.79 | 87.15 | 68.71 | 75.00 | 80.98 | 50.26 |
| | 2 | 66.13 | 89.19 | 71.13 | 75.00 | 83.57 | 47.75 |
| | 3 | 66.35 | 89.99 | 72.44 | 75.00 | 84.25 | 54.73 |
| | 4 | 65.17 | 89.89 | 72.18 | 75.01 | 83.99 | 50.78 |
| | 5 | 63.54 | 89.50 | 71.09 | 75.01 | 83.37 | 52.87 |
| | 6 | 57.84 | 86.44 | 66.42 | 75.00 | 77.01 | 42.72 |
| | 7 | 62.73 | 87.57 | 68.33 | 75.00 | 81.26 | 45.19 |
| | 8 | 66.30 | 89.07 | 71.55 | 75.00 | 83.61 | 52.95 |
| | 9 | 66.71 | 90.12 | 72.52 | 75.00 | 84.17 | 52.93 |
| | 10 | 65.29 | 89.44 | 71.62 | 75.00 | 83.81 | 54.83 |
| | 11 | 63.52 | 89.32 | 70.88 | 75.00 | 83.39 | 48.44 |
| | 12 | 59.49 | 86.41 | 66.45 | 75.00 | 77.98 | 50.64 |
| | 13 | 63.51 | 87.98 | 69.53 | 75.00 | 81.19 | 44.03 |
| | 14 | 67.14 | 89.40 | 72.20 | 75.01 | 83.80 | 47.97 |
| | 15 | 68.48 | 90.57 | 73.78 | 75.00 | 84.54 | 51.91 |
| | 16 | 68.27 | 90.34 | 73.63 | 75.01 | 84.48 | 50.11 |
| | 17 | 67.48 | 90.04 | 72.81 | 75.00 | 84.31 | 47.31 |
| | 18 | 66.44 | 89.80 | 72.02 | 75.00 | 84.35 | 49.94 |
| | 19 | 68.33 | 90.29 | 73.65 | 75.00 | 83.95 | 53.17 |
| | 20 | 68.13 | 90.67 | 73.85 | 75.00 | 84.61 | 54.20 |
| | 21 | 66.47 | 90.33 | 73.39 | 75.00 | 84.22 | 55.01 |
| | 22 | 59.62 | 90.53 | 73.63 | 75.00 | 84.61 | 50.53 |
| | 23 | 47.58 | 90.29 | 72.99 | 75.00 | 84.44 | 48.27 |
| | 24 | 32.27 | 90.29 | 73.96 | 75.00 | 84.42 | 51.33 |
| | 25 | 66.37 | 89.95 | 73.16 | 75.00 | 83.01 | 50.75 |
| | 26 | 67.96 | 90.65 | 73.13 | 75.00 | 83.94 | 52.29 |
| | 27 | 68.32 | 90.21 | 73.13 | 75.00 | 84.31 | 53.57 |
| | 28 | 68.48 | 90.57 | 73.78 | 75.00 | 84.54 | 51.91 |
| | 29 | 68.41 | 90.17 | 73.35 | 75.00 | 84.27 | 52.97 |
| | 30 | 68.12 | 89.67 | 72.53 | 75.01 | 84.27 | 50.47 |

Table S11: Top-1 on embeddings in all settings.

| Method | Run | ImageNet | iNat18 | Places205 | EuroSat | SUN397 | Cars |
|---|---|---|---|---|---|---|---|
| VICReg | 0 | 26.35 | 0.95 | 21.48 | 65.10 | 31.23 | 5.10 |
| | 1 | 30.54 | 1.39 | 21.07 | 63.80 | 31.60 | 5.24 |
| | 2 | 36.92 | 1.85 | 24.77 | 69.92 | 35.24 | 5.12 |
| | 3 | 47.60 | 4.34 | 35.48 | 87.84 | 47.62 | 7.56 |
| | 4 | 51.26 | 6.14 | 36.43 | 88.58 | 49.29 | 8.82 |
| | 5 | 54.39 | 7.87 | 38.04 | 88.44 | 51.88 | 9.76 |
| | 6 | 55.66 | 8.76 | 38.97 | 89.42 | 53.14 | 10.31 |
| | 7 | 56.33 | 9.56 | 39.65 | 89.76 | 53.39 | 10.53 |
| | 8 | 58.65 | 12.08 | 41.73 | 90.88 | 56.35 | 11.75 |
| | 9 | 59.71 | 13.47 | 42.72 | 91.28 | 57.26 | 11.95 |
| | 10 | 59.58 | 14.18 | 43.22 | 90.96 | 57.43 | 11.58 |
| | 11 | 59.22 | 14.63 | 43.48 | 91.34 | 57.75 | 11.99 |
| | 12 | 53.78 | 12.80 | 42.35 | 92.22 | 55.41 | 10.46 |
| | 13 | 57.94 | 14.36 | 43.61 | 91.60 | 57.89 | 11.30 |
| | 14 | 59.20 | 14.77 | 43.63 | 91.40 | 57.36 | 12.00 |
| | 15 | 59.73 | 14.20 | 43.31 | 91.66 | 57.04 | 11.95 |
| | 16 | 59.09 | 12.35 | 42.21 | 90.34 | 56.21 | 11.55 |
| | 17 | 58.23 | 11.34 | 41.02 | 90.16 | 54.97 | 11.69 |
| | 18 | 56.82 | 10.15 | 40.19 | 89.94 | 54.50 | 10.76 |
| | 19 | 55.22 | 9.26 | 39.02 | 90.00 | 53.07 | 10.98 |
| | 20 | 53.75 | 8.29 | 37.87 | 89.76 | 52.16 | 10.60 |
| | 21 | 51.53 | 12.97 | 40.84 | 91.64 | 54.26 | 13.13 |
| | 22 | 57.57 | 13.60 | 42.40 | 91.64 | 56.43 | 12.47 |
| | 23 | 59.71 | 13.47 | 42.72 | 91.28 | 57.26 | 11.95 |
| | 24 | 56.22 | 9.92 | 40.18 | 88.36 | 53.69 | 8.46 |
| | 25 | 36.22 | 2.48 | 29.59 | 85.06 | 42.46 | 7.55 |
| | 26 | 59.33 | 13.22 | 42.86 | 90.76 | 57.21 | 11.28 |
| | 27 | 59.51 | 13.37 | 42.69 | 91.34 | 56.66 | 11.53 |
| | 28 | 59.70 | 13.64 | 43.37 | 90.96 | 57.32 | 11.89 |
| | 29 | 59.03 | 14.00 | 43.10 | 91.50 | 57.44 | 12.27 |
| | 30 | 56.37 | 14.10 | 43.23 | 91.36 | 57.69 | 12.52 |
| | 31 | 49.96 | 12.36 | 41.93 | 91.52 | 57.13 | 11.37 |
| VICReg-exp | 0 | 58.19 | 12.56 | 41.93 | 91.82 | 57.13 | 10.19 |
| | 1 | 58.53 | 12.10 | 42.03 | 91.64 | 56.85 | 10.96 |
| | 2 | 57.41 | 10.78 | 40.89 | 90.44 | 55.24 | 10.73 |
| | 3 | 47.80 | 5.02 | 34.67 | 88.92 | 47.14 | 8.39 |
| | 4 | 25.14 | 1.02 | 22.04 | 77.22 | 32.66 | 5.17 |
| | 5 | 19.24 | 0.75 | 20.08 | 75.70 | 30.22 | 5.77 |
| | 6 | 12.03 | 0.51 | 16.95 | 70.18 | 26.06 | 4.15 |
| | 7 | 47.33 | 4.78 | 33.88 | 87.32 | 46.90 | 8.87 |
| | 8 | 53.72 | 7.74 | 38.04 | 89.46 | 52.37 | 9.76 |
| | 9 | 58.87 | 12.22 | 42.27 | 91.34 | 56.97 | 10.93 |
| | 10 | 58.09 | 12.70 | 42.56 | 91.58 | 56.90 | 10.42 |
| | 11 | 57.24 | 14.01 | 43.18 | 92.38 | 57.36 | 11.18 |

Table S12: Top-1 on embeddings in all settings, continued.

| Method | Run | ImageNet | iNat18 | Places205 | EuroSat | SUN397 | Cars |
|---|---|---|---|---|---|---|---|
| VICReg-ctr | 0 | 50.26 | 10.83 | 38.54 | 89.54 | 52.73 | 11.39 |
| | 1 | 50.99 | 9.81 | 38.43 | 90.06 | 53.90 | 10.53 |
| | 2 | 48.27 | 7.67 | 36.78 | 88.02 | 51.45 | 10.21 |
| | 3 | 36.77 | 3.20 | 30.01 | 84.12 | 43.85 | 6.68 |
| | 4 | 25.92 | 1.51 | 23.93 | 77.54 | 36.57 | 4.46 |
| | 5 | 17.69 | 0.70 | 18.19 | 69.00 | 28.30 | 3.73 |
| | 6 | 26.90 | 1.65 | 24.69 | 75.46 | 36.72 | 4.86 |
| | 7 | 44.31 | 5.81 | 34.82 | 87.84 | 49.23 | 9.25 |
| | 8 | 49.31 | 8.71 | 37.64 | 89.16 | 51.98 | 10.78 |
| | 9 | 46.43 | 8.38 | 36.11 | 89.44 | 50.27 | 10.25 |
| | 10 | 38.33 | 6.21 | 32.10 | 86.44 | 44.07 | 8.97 |
| SimCLR | 0 | 43.05 | 20.48 | 37.83 | 94.12 | 55.20 | 22.57 |
| | 1 | 49.69 | 21.28 | 41.82 | 93.48 | 58.23 | 21.38 |
| | 2 | 54.45 | 17.49 | 42.92 | 91.88 | 57.86 | 16.55 |
| | 3 | 50.24 | 8.36 | 39.22 | 88.42 | 51.94 | 11.58 |
| | 4 | 45.77 | 6.36 | 36.55 | 87.16 | 48.59 | 10.20 |
| | 5 | 41.14 | 4.81 | 34.04 | 84.76 | 45.36 | 9.10 |
| | 6 | 43.31 | 20.51 | 38.16 | 94.44 | 55.40 | 23.24 |
| | 7 | 49.60 | 21.51 | 41.93 | 93.88 | 59.05 | 22.43 |
| | 8 | 54.48 | 17.92 | 43.00 | 92.86 | 59.53 | 18.22 |
| | 9 | 50.72 | 8.65 | 39.64 | 89.44 | 54.47 | 12.96 |
| | 10 | 43.32 | 5.84 | 36.51 | 87.62 | 50.85 | 11.33 |
| | 11 | 41.15 | 5.02 | 34.26 | 85.52 | 48.16 | 10.87 |
| | 12 | 44.61 | 21.33 | 39.20 | 94.08 | 56.15 | 22.82 |
| | 13 | 51.54 | 21.50 | 42.64 | 94.40 | 59.87 | 21.30 |
| | 14 | 56.51 | 16.68 | 43.39 | 92.26 | 59.10 | 17.56 |
| | 15 | 56.89 | 10.35 | 41.21 | 90.38 | 56.37 | 14.04 |
| | 16 | 54.18 | 7.16 | 39.42 | 87.94 | 54.24 | 11.47 |
| | 17 | 49.19 | 4.98 | 37.12 | 87.14 | 50.85 | 10.33 |
| | 18 | 44.72 | 3.89 | 35.11 | 86.00 | 48.38 | 9.49 |
| | 19 | 57.06 | 10.12 | 41.18 | 89.52 | 56.20 | 13.19 |
| | 20 | 56.72 | 10.17 | 41.38 | 90.08 | 56.40 | 13.23 |
| | 21 | 56.14 | 10.26 | 41.47 | 89.12 | 56.74 | 13.18 |
| | 22 | 48.98 | 10.03 | 41.48 | 89.84 | 56.15 | 13.47 |
| | 23 | 35.92 | 10.03 | 41.33 | 89.74 | 56.38 | 13.87 |
| | 24 | 28.26 | 9.68 | 41.22 | 86.88 | 56.02 | 13.13 |
| | 25 | 52.65 | 9.98 | 40.47 | 90.20 | 55.38 | 13.72 |
| | 26 | 55.98 | 9.88 | 40.84 | 89.00 | 55.43 | 13.41 |
| | 27 | 56.43 | 10.03 | 41.04 | 89.60 | 56.23 | 13.89 |
| | 28 | 56.89 | 10.35 | 41.21 | 90.38 | 56.37 | 14.04 |
| | 29 | 56.65 | 10.30 | 41.40 | 89.22 | 56.42 | 13.79 |
| | 30 | 56.56 | 10.60 | 41.61 | 90.14 | 56.82 | 13.90 |

Table S13: Top-1 on embeddings in all settings, continued.

| Method | Run | ImageNet | CIFAR10 | CIFAR100 | FOOD101 | VOC07 | CLEVR-count |
|---|---|---|---|---|---|---|---|
| VICReg | 0 | 26.35 | 59.23 | 25.84 | 75.00 | 66.15 | 21.75 |
| | 1 | 30.54 | 60.57 | 25.67 | 75.01 | 67.33 | 19.64 |
| | 2 | 36.92 | 63.77 | 29.67 | 75.00 | 70.94 | 23.72 |
| | 3 | 47.60 | 75.59 | 44.58 | 75.01 | 75.90 | 41.44 |
| | 4 | 51.26 | 76.88 | 45.26 | 75.01 | 76.85 | 34.20 |
| | 5 | 54.39 | 78.34 | 49.63 | 75.01 | 77.85 | 40.24 |
| | 6 | 55.66 | 78.71 | 49.89 | 75.01 | 78.17 | 38.37 |
| | 7 | 56.33 | 78.89 | 50.52 | 75.01 | 78.41 | 39.88 |
| | 8 | 58.65 | 79.57 | 50.95 | 75.01 | 79.46 | 43.13 |
| | 9 | 59.71 | 80.43 | 52.75 | 75.01 | 79.50 | 43.93 |
| | 10 | 59.58 | 80.59 | 53.80 | 75.02 | 79.19 | 43.87 |
| | 11 | 59.22 | 80.94 | 53.66 | 75.03 | 78.90 | 44.90 |
| | 12 | 53.78 | 78.96 | 51.83 | 75.03 | 76.17 | 43.71 |
| | 13 | 57.94 | 81.43 | 53.92 | 75.02 | 78.04 | 45.23 |
| | 14 | 59.20 | 81.04 | 53.88 | 75.03 | 79.18 | 43.75 |
| | 15 | 59.73 | 81.16 | 53.35 | 75.02 | 79.18 | 44.39 |
| | 16 | 59.09 | 80.46 | 52.82 | 75.02 | 79.20 | 45.92 |
| | 17 | 58.23 | 79.76 | 51.77 | 75.01 | 79.58 | 36.53 |
| | 18 | 56.82 | 79.20 | 51.10 | 75.01 | 78.62 | 37.81 |
| | 19 | 55.22 | 78.82 | 50.19 | 75.01 | 78.36 | 38.50 |
| | 20 | 53.75 | 77.87 | 49.34 | 75.01 | 78.07 | 38.62 |
| | 21 | 51.53 | 78.45 | 51.46 | 75.01 | 75.99 | 49.13 |
| | 22 | 57.57 | 80.67 | 52.87 | 75.02 | 78.53 | 45.93 |
| | 23 | 59.71 | 80.43 | 52.75 | 75.01 | 79.50 | 43.93 |
| | 24 | 56.22 | 75.80 | 45.73 | 75.02 | 78.90 | 39.71 |
| | 25 | 36.22 | 72.55 | 41.50 | 75.00 | 73.12 | 41.73 |
| | 26 | 59.33 | 79.58 | 52.25 | 75.02 | 79.61 | 44.89 |
| | 27 | 59.51 | 80.26 | 52.55 | 75.01 | 79.29 | 42.99 |
| | 28 | 59.70 | 79.74 | 52.92 | 75.02 | 79.67 | 46.11 |
| | 29 | 59.03 | 81.25 | 54.96 | 75.01 | 80.10 | 43.33 |
| | 30 | 56.37 | 80.81 | 53.55 | 75.01 | 80.11 | 46.97 |
| | 31 | 49.96 | 80.86 | 53.10 | 75.01 | 79.68 | 47.09 |
| VICReg-exp | 0 | 58.19 | 80.80 | 52.30 | 75.01 | 79.73 | 45.89 |
| | 1 | 58.53 | 80.15 | 53.08 | 75.01 | 80.18 | 43.75 |
| | 2 | 57.41 | 79.22 | 51.69 | 75.01 | 79.39 | 43.92 |
| | 3 | 47.80 | 76.70 | 45.93 | 75.00 | 76.21 | 43.52 |
| | 4 | 25.14 | 66.91 | 33.49 | 75.00 | 66.12 | 37.21 |
| | 5 | 19.24 | 65.85 | 29.87 | 75.00 | 62.04 | 34.52 |
| | 6 | 12.03 | 62.48 | 26.06 | 75.00 | 55.71 | 34.17 |
| | 7 | 47.33 | 76.91 | 46.47 | 75.00 | 76.48 | 41.40 |
| | 8 | 53.72 | 77.99 | 48.65 | 75.00 | 78.72 | 44.80 |
| | 9 | 58.87 | 80.36 | 53.78 | 75.01 | 80.15 | 43.85 |
| | 10 | 58.09 | 80.64 | 53.47 | 75.01 | 79.98 | 45.68 |
| | 11 | 57.24 | 81.10 | 54.28 | 75.01 | 79.58 | 43.71 |

Table S14: Top-1 on embeddings in all settings, continued.

| Method | Run | ImageNet | CIFAR10 | CIFAR100 | FOOD101 | VOC07 | CLEVR-count |
|--------|-----|----------|---------|----------|---------|-------|-------------|
| VICReg-ctr | 0 | 50.26 | 78.76 | 49.34 | 75.00 | 77.89 | 37.61 |
| | 1 | 50.99 | 78.63 | 49.80 | 75.00 | 78.77 | 43.76 |
| | 2 | 48.27 | 77.86 | 48.75 | 75.00 | 78.48 | 40.49 |
| | 3 | 36.77 | 73.83 | 40.44 | 75.00 | 72.19 | 38.16 |
| | 4 | 25.92 | 66.81 | 32.82 | 75.00 | 65.32 | 31.63 |
| | 5 | 17.69 | 63.94 | 25.50 | 75.00 | 58.48 | 31.40 |
| | 6 | 44.31 | 76.82 | 46.43 | 75.00 | 77.16 | 42.48 |
| | 7 | 49.31 | 77.70 | 48.66 | 75.00 | 78.50 | 40.81 |
| | 8 | 46.43 | 75.51 | 46.54 | 75.00 | 76.97 | 44.25 |
| | 9 | 38.33 | 71.51 | 40.68 | 75.00 | 72.67 | 41.31 |
| SimCLR | 0 | 43.05 | 75.84 | 52.39 | 75.00 | 71.43 | 46.86 |
| | 1 | 49.69 | 78.32 | 53.20 | 75.00 | 76.19 | 50.11 |
| | 2 | 54.45 | 79.24 | 51.56 | 75.00 | 78.10 | 49.63 |
| | 3 | 50.24 | 78.36 | 47.70 | 75.00 | 79.07 | 46.29 |
| | 4 | 45.77 | 75.73 | 44.37 | 75.00 | 77.46 | 46.63 |
| | 5 | 41.14 | 74.04 | 41.88 | 75.00 | 75.03 | 45.47 |
| | 6 | 43.31 | 75.66 | 53.31 | 75.00 | 71.37 | 41.40 |
| | 7 | 49.60 | 78.36 | 54.79 | 75.00 | 76.47 | 46.46 |
| | 8 | 54.48 | 80.39 | 55.39 | 75.00 | 78.38 | 49.53 |
| | 9 | 50.72 | 79.32 | 51.01 | 75.00 | 79.27 | 42.25 |
| | 10 | 43.32 | 76.43 | 48.07 | 75.00 | 77.59 | 44.67 |
| | 11 | 41.15 | 74.75 | 45.06 | 75.00 | 74.79 | 46.56 |
| | 12 | 44.61 | 76.97 | 53.29 | 75.00 | 72.71 | 47.15 |
| | 13 | 51.54 | 79.20 | 56.45 | 75.00 | 76.80 | 37.03 |
| | 14 | 56.51 | 79.48 | 53.99 | 75.00 | 78.55 | 43.05 |
| | 15 | 56.89 | 79.35 | 52.58 | 75.00 | 79.73 | 43.51 |
| | 16 | 54.18 | 78.17 | 49.62 | 75.00 | 79.71 | 43.32 |
| | 17 | 49.19 | 76.59 | 46.98 | 75.00 | 78.61 | 43.48 |
| | 18 | 44.72 | 76.68 | 45.62 | 74.99 | 76.95 | 42.88 |
| | 19 | 57.06 | 79.83 | 52.19 | 75.00 | 79.73 | 45.97 |
| | 20 | 56.72 | 79.41 | 53.12 | 75.00 | 80.05 | 45.65 |
| | 21 | 56.14 | 79.46 | 52.25 | 75.00 | 79.64 | 47.19 |
| | 22 | 48.98 | 79.39 | 52.28 | 75.00 | 79.89 | 45.92 |
| | 23 | 35.92 | 79.52 | 52.15 | 75.00 | 79.74 | 43.86 |
| | 24 | 28.26 | 79.28 | 51.25 | 75.00 | 79.90 | 45.39 |
| | 25 | 52.65 | 79.30 | 52.85 | 75.00 | 78.74 | 42.71 |
| | 26 | 55.98 | 79.51 | 52.21 | 75.00 | 79.34 | 44.75 |
| | 27 | 56.43 | 78.52 | 52.04 | 75.00 | 79.55 | 45.19 |
| | 28 | 56.89 | 79.35 | 52.58 | 75.00 | 79.73 | 43.51 |
| | 29 | 56.65 | 78.98 | 51.80 | 75.00 | 79.82 | 43.69 |
| | 30 | 56.56 | 78.36 | 51.81 | 74.99 | 79.78 | 44.50 |

Table S15: Rank after projector in all settings.

| Method | Run | ImageNet | iNat18 | Places205 | EuroSat | SUN397 | Cars |
|---|---|---|---|---|---|---|---|
| VICReg | 0 | 102.07 | 38.10 | 44.39 | 14.61 | 32.40 | 7.03 |
| | 1 | 229.81 | 92.53 | 129.47 | 88.78 | 98.44 | 12.58 |
| | 2 | 374.25 | 135.79 | 206.29 | 120.31 | 163.31 | 19.77 |
| | 3 | 612.12 | 261.34 | 336.16 | 228.60 | 265.64 | 38.90 |
| | 4 | 831.49 | 382.55 | 467.68 | 366.78 | 374.50 | 59.15 |
| | 5 | 952.55 | 449.44 | 539.24 | 428.87 | 435.94 | 77.36 |
| | 6 | 1033.93 | 493.50 | 587.19 | 477.69 | 478.34 | 88.28 |
| | 7 | 1088.13 | 531.16 | 630.80 | 514.70 | 517.47 | 99.97 |
| | 8 | 1442.63 | 726.28 | 849.29 | 693.16 | 723.53 | 161.76 |
| | 9 | 1809.06 | 947.81 | 1110.80 | 855.76 | 954.83 | 210.06 |
| | 10 | 1920.81 | 1054.70 | 1247.93 | 870.56 | 1075.89 | 258.33 |
| | 11 | 1938.44 | 1087.45 | 1275.60 | 924.66 | 1119.33 | 306.90 |
| | 12 | 1937.78 | 1100.54 | 1337.88 | 963.14 | 1172.38 | 382.18 |
| | 13 | 1944.95 | 1095.95 | 1307.62 | 968.96 | 1155.65 | 352.50 |
| | 14 | 1940.04 | 1095.91 | 1280.85 | 910.16 | 1126.89 | 324.51 |
| | 15 | 1942.12 | 1049.72 | 1240.87 | 893.25 | 1070.12 | 269.96 |
| | 16 | 1521.07 | 782.39 | 919.54 | 725.49 | 771.86 | 169.75 |
| | 17 | 1278.67 | 637.18 | 757.19 | 606.98 | 627.48 | 128.96 |
| | 18 | 1079.67 | 532.00 | 634.88 | 527.59 | 524.80 | 111.28 |
| | 19 | 909.71 | 446.52 | 525.65 | 454.22 | 431.44 | 88.55 |
| | 20 | 777.82 | 376.39 | 447.53 | 378.06 | 360.41 | 73.57 |
| | 21 | 1409.29 | 890.97 | 996.12 | 814.00 | 889.66 | 352.57 |
| | 22 | 1652.41 | 936.47 | 1070.40 | 837.76 | 932.17 | 275.04 |
| | 23 | 1809.06 | 947.81 | 1110.80 | 855.76 | 954.83 | 210.06 |
| | 24 | 1422.16 | 648.60 | 813.33 | 532.92 | 650.33 | 91.44 |
| | 25 | 101.29 | 44.12 | 46.00 | 20.77 | 36.60 | 10.68 |
| | 26 | 1821.80 | 959.98 | 1130.27 | 840.12 | 962.04 | 221.58 |
| | 27 | 1814.64 | 948.47 | 1107.25 | 856.12 | 946.73 | 218.36 |
| | 28 | 1728.89 | 913.31 | 1065.74 | 814.04 | 911.39 | 216.25 |
| | 29 | 1587.36 | 859.56 | 1008.93 | 807.57 | 864.18 | 244.56 |
| | 30 | 1384.68 | 757.81 | 881.53 | 716.36 | 767.14 | 229.93 |
| | 31 | 974.91 | 508.81 | 613.44 | 508.01 | 526.43 | 143.61 |
| VICReg-exp | 0 | 1006.58 | 530.95 | 637.48 | 501.16 | 551.60 | 142.88 |
| | 1 | 1002.17 | 521.34 | 626.39 | 515.00 | 534.56 | 132.72 |
| | 2 | 922.59 | 473.26 | 564.18 | 475.88 | 472.06 | 119.75 |
| | 3 | 399.09 | 192.27 | 233.31 | 202.71 | 189.78 | 36.95 |
| | 4 | 63.82 | 30.25 | 36.98 | 21.39 | 30.63 | 7.90 |
| | 5 | 19.47 | 12.49 | 9.57 | 6.33 | 7.96 | 3.58 |
| | 6 | 9.42 | 7.19 | 5.41 | 3.80 | 4.73 | 2.55 |
| | 7 | 375.38 | 180.63 | 216.71 | 191.99 | 176.94 | 31.86 |
| | 8 | 636.60 | 314.20 | 380.13 | 341.21 | 312.04 | 66.31 |
| | 9 | 1002.29 | 528.76 | 629.07 | 517.84 | 536.91 | 139.28 |
| | 10 | 1048.58 | 556.24 | 673.46 | 547.15 | 581.30 | 158.24 |
| | 11 | 1326.31 | 733.86 | 875.62 | 707.34 | 771.39 | 208.83 |

Table S16: Rank after projector in all settings, continued.

| Method | Run | ImageNet | iNat18 | Places205 | EuroSat | SUN397 | Cars |
|---|---|---|---|---|---|---|---|
| VICReg-ctr | 0 | 382.33 | 224.68 | 252.33 | 207.81 | 220.68 | 69.48 |
| | 1 | 278.88 | 163.91 | 183.32 | 154.70 | 160.71 | 50.29 |
| | 2 | 169.33 | 101.44 | 114.49 | 97.89 | 99.84 | 34.97 |
| | 3 | 48.47 | 32.38 | 34.93 | 32.77 | 31.76 | 12.53 |
| | 4 | 23.22 | 16.72 | 17.90 | 17.70 | 16.63 | 7.38 |
| | 5 | 12.88 | 10.03 | 10.31 | 10.66 | 9.71 | 5.01 |
| | 6 | 22.96 | 16.87 | 17.77 | 17.30 | 16.55 | 7.61 |
| | 7 | 96.33 | 62.08 | 68.05 | 60.68 | 60.39 | 22.59 |
| | 8 | 251.52 | 146.09 | 166.32 | 138.75 | 143.81 | 45.73 |
| | 9 | 309.22 | 177.32 | 204.38 | 170.65 | 175.81 | 53.83 |
| | 10 | 316.89 | 184.83 | 213.74 | 175.10 | 185.91 | 59.07 |
| SimCLR | 0 | 109.07 | 105.65 | 104.65 | 76.13 | 105.64 | 92.59 |
| | 1 | 164.07 | 148.71 | 149.89 | 100.17 | 148.00 | 113.61 |
| | 2 | 244.34 | 184.32 | 203.04 | 129.53 | 188.30 | 105.89 |
| | 3 | 150.90 | 94.61 | 116.94 | 83.98 | 102.17 | 40.64 |
| | 4 | 87.69 | 57.78 | 67.23 | 54.62 | 59.79 | 25.36 |
| | 5 | 63.68 | 42.23 | 48.22 | 40.83 | 43.22 | 18.41 |
| | 6 | 110.59 | 106.83 | 105.83 | 76.97 | 106.82 | 93.98 |
| | 7 | 165.49 | 149.55 | 150.27 | 103.19 | 148.65 | 113.60 |
| | 8 | 246.56 | 184.69 | 204.24 | 128.96 | 189.86 | 107.43 |
| | 9 | 164.66 | 102.61 | 128.12 | 95.47 | 112.20 | 43.29 |
| | 10 | 9.88 | 30.27 | 2.74 | 55.46 | 65.08 | 25.57 |
| | 11 | 63.61 | 42.00 | 48.40 | 40.86 | 43.20 | 18.62 |
| | 12 | 122.60 | 118.57 | 116.93 | 85.13 | 118.16 | 103.25 |
| | 13 | 197.36 | 173.50 | 176.32 | 116.61 | 173.24 | 128.89 |
| | 14 | 313.67 | 220.05 | 239.53 | 160.52 | 222.80 | 111.73 |
| | 15 | 299.47 | 172.75 | 209.43 | 140.66 | 183.51 | 61.44 |
| | 16 | 220.63 | 122.46 | 150.73 | 106.96 | 130.16 | 40.02 |
| | 17 | 128.33 | 71.75 | 90.40 | 65.77 | 78.64 | 26.24 |
| | 18 | 71.75 | 48.95 | 64.25 | 48.65 | 54.84 | 18.93 |
| | 19 | 301.92 | 173.11 | 211.03 | 147.45 | 185.04 | 60.83 |
| | 20 | 299.75 | 173.05 | 208.52 | 141.84 | 182.21 | 61.56 |
| | 21 | 299.96 | 173.61 | 209.18 | 144.25 | 181.99 | 61.11 |
| | 22 | 300.90 | 173.89 | 209.47 | 147.78 | 184.45 | 61.40 |
| | 23 | 300.58 | 174.18 | 207.19 | 142.58 | 184.29 | 60.94 |
| | 24 | 300.83 | 174.63 | 207.18 | 146.11 | 182.27 | 60.50 |
| | 25 | 11.56 | 15.95 | 31.99 | 144.87 | 13.55 | 3.92 |
| | 26 | 293.13 | 172.80 | 211.66 | 139.57 | 184.94 | 65.02 |
| | 27 | 295.23 | 173.07 | 208.46 | 139.91 | 181.05 | 62.32 |
| | 28 | 299.47 | 172.75 | 209.43 | 140.66 | 183.51 | 61.44 |
| | 29 | 298.69 | 172.12 | 206.63 | 142.92 | 181.39 | 60.88 |
| | 30 | 294.42 | 170.26 | 201.98 | 141.29 | 177.39 | 58.94 |

