# OpenReview forum: "RankMe: Assessing the Downstream Performance of Pretrained Self-Supervised Representations by Their Rank"
_ICLR.cc/2023/Conference — Submitted to ICLR 2023_

### Official Review · Reviewer_NJ5U · 2022-10-23

**Confidence:** 4
**Correctness:** 2
**Technical Novelty And Significance:** 2
**Empirical Novelty And Significance:** 2
**Recommendation:** 3

**Clarity, Quality, Novelty And Reproducibility:**

The writing of the paper is fairly clear and the problem being studied is an important one. My concerns are primarily about the quality of the experimental analysis and the utility of the proposed method, which I have discussed above.

**Strength And Weaknesses:**

I find the premise of the paper—selecting hyperparameters for SSL methods without labels—compelling, particularly given the rise of large vision/language models trained on massive amounts of data. However, both conceptually, and experimentally, the effectiveness of the proposed approach is not clear.

[Conceptually] There are several leaps from the setting in which the first equation in S3.1 holds to practice: (i) regression to classification, (ii) the linear probe does not overfit, (iii) embedding rank scales monotonically across datasets and (iv) embeddings and representations are monotonically linked. It is not clear that these hold true in practice. For instance, depending on the amount of data available for the target task, overfitting of the linear probe is quite likely in practice. Additionally, for many natural transfer tasks, it is possible that the embedding rank on the source dataset does not perfectly correlate with that on the target.

[Experimentally] Even in the ablations conducted by the authors, there are several instances where practice doesn’t line up with the authors intuitions (eg, Figures 2, 3 and 4). As a result, it is hard to assess how reliably this method will work as the source dataset is varied or we consider a larger assortment of downstream tasks. For instance, the assumptions do not hold on Stanford Cars, where rankme also fails to perform good model selection (Table S3).

The results would be a lot more compelling if the authors considered: (i) models trained on a different source dataset other than ImageNet (eg, the CLIP vision encoders from [https://github.com/mlfoundations/open_clip](https://github.com/mlfoundations/open_clip)), and (ii) more downstream tasks (eg, all the tasks from Kornblith et al. [arXiv:1805.08974]).

Another drawback of this approach (compared to the oracle of training a linear probe on ImageNet) is that it cannot be used to contrast models trained with different objectives/architectures, which is arguably the bigger consideration in practice. For instance, in Figure 3, on  every dataset, there are points corresponding to SimCLR and VICReg with the same test performance but very different rankme score.

Other comments/questions:

- What was the motivation for the four selected downstream tasks considered in the paper? Given that the average results that the authors report are entirely tied to the choice of datasets, the authors should expand their selection to include more standard tasks. The numbers reported in the paper should also be per task rather than the average.
- How is the linear probe fit and how are its hyperparameters selected?

**Summary Of The Paper:**

This paper proposes a new methodology for tuning hyperparameters of SSL methods, without using labels. The method is motivated by a result in the linear regression setting wherein the train accuracy of a linear probe improves with embedding rank. Based on this, the authors propose that a robust rank measure (originally introduced in prior work) that can be used for model selection.

**Summary Of The Review:**

Overall, I am leaning towards rejection because the utility of the proposed approach seems to be fairly limited (hyperparameter selection for a single objective+architecture). Even in that setting, the current experimental results are not sufficiently convincing. I would be willing to increase my score if the authors expanded their analysis to include: (i) models trained on a different source dataset other than ImageNet (eg, the CLIP vision encoders from https://github.com/mlfoundations/open_clip), and (ii) more downstream tasks (eg, all the tasks from Kornblith et al. [arXiv:1805.08974]).

---

> ### Author Response · Authors · 2022-11-16
> **Response to review 1/2**
>
> > There are several leaps from the setting in which the first equation in S3.1 holds to practice: (i) regression to classification, (ii) the linear probe does not overfit, (iii) embedding rank scales monotonically across datasets and (iv) embeddings and representations are monotonically linked. It is not clear that these hold true in practice. For instance, depending on the amount of data available for the target task, overfitting of the linear probe is quite likely in practice. Additionally, for many natural transfer tasks, it is possible that the embedding rank on the source dataset does not perfectly correlate with that on the target
>
> We entirely agree with the reviewer and in fact those assumptions, required for RankMe to be “provably meaningful” can easily be seen as too restrictive. We have made sure to tame down our claims throughout the paper, only using those as insights and possible explanations instead of framing RankMe as a provable method. We believe however that our clear statement of those assumptions already indicate the scenarios in which RankMe would naturally fail e.g. (as you pointed out) if the downstream task as very small training set (preventing (ii)) or if the target data distribution is very far form the source one (probably preventing (iii)).
>
> >[Experimentally] Even in the ablations conducted by the authors, there are several instances where practice doesn’t line up with the authors intuitions (eg, Figures 2, 3 and 4). As a result, it is hard to assess how reliably this method will work as the source dataset is varied or we consider a larger assortment of downstream tasks. For instance, the assumptions do not hold on Stanford Cars, where rankme also fails to perform good model selection (Table S3).
>
> We agree that further experiments are required to provide a complete assessment of the method. We would still like to point out that this is the largest study of dimensional collapse to date, and that the current results are already extremely computationally expensive.
> The failure on StanfordCars brings valuable information as the lack of semantic overlap is a failure mode that we describe in the paper, which is here validated empirically and not based solely on intuition.
>
>
> Nonetheless, to improve the quality of our experiments, we have added 5 new datasets: VOC07, CIFAR10,CIFAR100, Food101 and CLEVR-count. See supplementary section D for the visualizations. The numerical results are included in the performance reported in the main body.
>
>
> Similarly, we also validated RankMe on iNaturalist pretraining (see supplementary section B), albeit at a smaller scale than our previous experiments due to time constraints. There we even see that for downstream performance on ImageNet the iNaturalist labels provide a suboptimal oracle compared to RankMe.
>
> >Another drawback of this approach (compared to the oracle of training a linear probe on ImageNet) is that it cannot be used to contrast models trained with different objectives/architectures, which is arguably the bigger consideration in practice. For instance, in Figure 3, on every dataset, there are points corresponding to SimCLR and VICReg with the same test performance but very different rankme score.
>
> We thank the reviewer for raising this concern. In fact we have made sure to mention throughout the paper that RankMe is only for hyper-parameter selection within the same method/model. Note that although this is indeed a strong limitation, it is also the case of other methods that attempt to provide label-free quality metric of a model’s representation. We hope that this limitation can be the base of future work. As mentioned by the reviewer, oracle linear probe would be able to compare architectures naturally, but at the cost of requiring labels which depending on the situation may not be preferable, or even possible.
>
> >What was the motivation for the four selected downstream tasks considered in the paper? Given that the average results that the authors report are entirely tied to the choice of datasets, the authors should expand their selection to include more standard tasks. The numbers reported in the paper should also be per task rather than the average.
>
> While we previously provided a discussion in section 3.2 about why we chose certain datasets, we have also added 5 new tasks: VOC07, CIFAR10,CIFAR100, Food101 and CLEVR-count. This should provide a wider variety of standard tasks.
> To summarize, we chose datasets that provided settings that differed from ImageNet by their fine grained nature (iNaturalist), non object-centric properties (SUN397), different domain (EuroSat) or limited semantic overlap (StanfordCars).
> The numbers were reported per task in supplementary section J, which we have extended with the new datasets.

---

> > ### Author Response · Authors · 2022-11-16
> > **Response to review 2/2**
> >
> > > How is the linear probe fit and how are its hyperparameters selected?
> >
> > The details are available in supplementary section I which we have updated to include the new datasets. This choice was made due to space limitations. We relied on the default parameters of VISSL(https://github.com/facebookresearch/vissl) when it gave performance that made sense, i.e. not too far from the performance reported in the literature, and tuned the learning rate if necessary. The general protocol is to use SGD with momentum and a step lr rule to reduce it during training.
> >
> > >I would be willing to increase my score if the authors expanded their analysis to include: (i) models trained on a different source dataset other than ImageNet (eg, the CLIP vision encoders from https://github.com/mlfoundations/open_clip), and (ii) more downstream tasks (eg, all the tasks from Kornblith et al. [arXiv:1805.08974]).
> >
> > We have added experiments with iNaturalist18 pretraining, on which we applied our model selection approach to find SimCLR's temperature and VICReg's covariance weight. The details are available in supplementary section B. We see that RankMe offers a similar level of performance as on ImageNet pretrainings, even beating the iNaturalist oracle when evaluating on ImageNet.
> >
> > As previously mentioned, per your suggestion, we have added 5 new standard datasets for downstream evaluation: CIFAR10, CIFAR100, VOC07, Food101 and CLEVR-cnt.
> >
> > We opted for the datasets that would give us meaningful information, with a restricted computational budget due to the cost of evaluating more than 80 models.
> > We did not include Flowers or Dogs, as they are already represented by a dataset such as iNaturalist. CLEVR-cnt was not included in Kornblith et al. but provides an interesting task of object counting, going beyond simple image classification. We hope that those additional set of experiments help further validate RankMe and answer the reviewer’s concerns.

---

> > ### Comment · Reviewer_NJ5U · 2022-11-28
> > **Post-rebuttal update**
> >
> > I thank the authors for their detailed response and for running additional evaluations. I do however still feel that the experimental results are not entirely convincing---including those in Table S3 and S4 on the new target datasets. It seems that the performance of RankMe is comparable to the α-ReQ baseline in most cases. The study on DINO is very limited (only in-distribution, no comparison to α-ReQ). And further, as I said in my review, comparing different models (architectures, trained on different datasets) seems like the bigger challenge during model selection than the setting considered in the paper. I am thus inclined to keep my score as is.

---

> > > ### Author Response · Authors · 2022-12-07
> > > **Response to update**
> > >
> > > The main issue with the datasets used for out of distribution evaluation is that some of them are significantly simpler than (or are even semantic subsets of) the pretraining dataset here, meaning that there is not much insights that can be gained by evaluating on them. It is reasonable to expect that the best performing model on ImageNet will also be the best on CIFAR-10, up to the randomness induced by the training of the evaluation head.
> > >
> > > The performance of RankMe and $\alpha$-ReQ are similar, but we want to emphasize that $\alpha$-ReQ suffers from significantly bigger worst-case performance drops. For example for the temperature of SimCLR, it suffers from a drop of 3.6 points compared to RankMe, and 5 points compared to the ImageNet oracle.
> > >
> > > The study on DINO is currently limited since we preferred to focus on the experiments that you originally suggested. It is also not meant to be definitive by any means and is here as an insight into expanding our experimental results.
> > >
> > > While we agree that comparison in more diverse settings is a bigger challenge, tackling the problem of unsupervised evaluation inside a given method is still not solved and is already challenging, as we hope to have shown. This is a natural next step that requires solid foundations in a simpler, yet already useful setting.

---

### Official Review · Reviewer_Sb8P · 2022-10-24

**Confidence:** 3
**Correctness:** 3
**Technical Novelty And Significance:** 2
**Empirical Novelty And Significance:** 2
**Recommendation:** 3

**Clarity, Quality, Novelty And Reproducibility:**

Overall, the clarity and quality are low and should be improved.

The novelty is limited as discussed above.

The reproducibility of this work may be good.


**Strength And Weaknesses:**

# Strength
- The introduced method is simple, computationally friendly, and effective.
- Extensive experiments have proved the correctness of RankMe.
- The idea of using rank as a metric to select hyperparameters is interesting. Its competitiveness with traditional oracle-based hyperparameter selection methods makes it a promising tool.

# Weaknesses
- The novelty of this work is limited. It is known that self-supervised learning suffers from dimensional collapse. The rank of the embedding is a measure of the degree of dimensional collapse, so it can reflect the downstream performance to some degree. The main result Eq 2 comes from the empirical observation, which is pretty not rigorous (but the authors have claimed things like principled guidances and theoretically motivated balabala).
- The writing, presentation, and organization of the paper are poor. There are many grammar errors, such as, in page 1, INTRODUCTION, “tuning JE-SSL methods on unlabeled datasets remain challenging” would be “tuning JE-SSL methods on unlabeled datasets remains challenging”. And There are typos, e.g., in page 3, Fig.1, “selwction” would be “selection”. The derivation of RankMe in section 3.1 is a bit difficult to follow and sometimes obscure. Please check the manuscript carefully.
- RankMe relies on multiple hypotheses. Though the authors have validated them with some studies, they should also clarify when/where/how these hypotheses apply and fail so that the followers know the applicability of RankMe.
- RankMe can only be used to tune hyperparameters but not for model/method selection? We know cross-validation is indeed a canonical method for hyperparameter tuning and model/method selection. Why/How RankMe is better than cross-validation? In the aspect of performance of applicability?

## Minor issues
- In Figure 1, the lines are emphasized by deliberately ignoring the outliers. I cannot see a strong correlation when separately inspecting the red points (corresponding to SimCLR).


**Summary Of The Paper:**

This paper proposes RankMe, a method that can assess the quality of JE-SSL representations and work as a predictor of the representations’ performance on target datasets. Without requiring any labels, training, or parameters to tune, this method is simple and computationally friendly. Furthermore, this paper demonstrates that RankMe can be used for hyperparameter selection.

**Summary Of The Review:**

Given the poor writing, limited novelty, and some technical issues on hypotheses, I am learning to reject this paper at this time.

---

> ### Author Response · Authors · 2022-11-16
> **Response to review**
>
> > The novelty of this work is limited. It is known that self-supervised learning suffers from dimensional collapse. The rank of the embedding is a measure of the degree of dimensional collapse, so it can reflect the downstream performance to some degree.
>
> We thank the reviewer for raising this concern. First, we entirely agree with the reviewer that dimensional collapse is a known phenomenon, see eg [1,2] that we already referenced in our submission. That being said, our core contribution is not on that front, but rather on providing a thorough empirical validation that the rank alone is in fact well correlated with downstream performances.
> No previous study of dimensional collapse was done at our scale and in settings that are as comparable as ours.
>
>
> [1] Jing, Li, et al. "Understanding dimensional collapse in contrastive self-supervised learning." arXiv preprint arXiv:2110.09348 (2021)
>
> [2]Hua, Tianyu, et al. "On feature decorrelation in self-supervised learning." Proceedings of the IEEE/CVF International Conference on Computer Vision. 2021.
>
> > The main result Eq 2 comes from the empirical observation, which is pretty not rigorous (but the authors have claimed things like principled guidances and theoretically motivated balabala).
>
> The construction of RankMe combined theoretically grounded results and empirical ones which remain realistic in most settings e.g. little gap between train and test accuracies (realistic with linear classifier and large datasets). In the revised manuscript, we have made sure to (i) explicit those assumptions more strongly, and (ii) tame down throughout the paper that rank me is theoretically grounded.
>
> > The writing, presentation, and organization of the paper are poor. There are many grammar errors, such as, in page 1, INTRODUCTION, “tuning JE-SSL methods on unlabeled datasets remain challenging” would be “tuning JE-SSL methods on unlabeled datasets remains challenging”. And There are typos, e.g., in page 3, Fig.1, “selwction” would be “selection”. The derivation of RankMe in section 3.1 is a bit difficult to follow and sometimes obscure. Please check the manuscript carefully.
>
> We thank the reviewer for their careful reading and raising those writing mistakes. We have made sure to correct the aforementioned ones, and we have done multiple passes through the manuscript to streamline the presentation.
>
> > RankMe relies on multiple hypotheses. Though the authors have validated them with some studies, they should also clarify when/where/how these hypotheses apply and fail so that the followers know the applicability of RankMe.
>
> We entirely agree with the reviewer, and following our answer to the reviewer’s second comment, we have made sure to update the manuscript to explicit RankMe’s reliance on those hypotheses. The limitations of the hypothesis are also described in section 3.1.
>
>  Regarding empirical validation, we have done hundreds of studies that we hope should be representative enough, we have especially added more downstream tasks and SSL methods to that end in the revised manuscript (e.g. DINO method, INaturalist source dataset, and CIFAR10/100,Food101,CLEVR-count target datasets).
>
> > RankMe can only be used to tune hyperparameters but not for model/method selection? We know cross-validation is indeed a canonical method for hyperparameter tuning and model/method selection. Why/How RankMe is better than cross-validation? In the aspect of performance of applicability?
>
> As we briefly summarized in the general answer, RankMe differs from standard cross-validation in that we do not require any labels. Now if one were to take RankMe’s score as the performance metric, then cross-validation could be applied jointly with RankMe to select a model’s hyper-parameter, as we have illustrated in Fig 5. Being able to compare across methods would indeed be a desired property of RankMe but this limitation is also present in other methods. Due to the great benefit of doing cross-validation label-free (even if within the same architecture) we believe that this limitation should be seen as a future work opportunity.
>
> Regarding applicability, the fact that RankMe does not rely on labels makes it more general than the classical approach, especially for large unlabeled datasets or datasets where labels are expensive to obtain.
>
> >In Figure 1, the lines are emphasized by deliberately ignoring the outliers. I cannot see a strong correlation when separately inspecting the red points (corresponding to SimCLR).
>
> The lines are present to show the upper envelope of performance, in order to highlight the necessary condition of a high rank. For SimCLR, the point cloud is harder to analyze since we also have results with suboptimal parameters that achieve lower performance. In Fig 5 however we can see more clearly what happens when the temperature changes which should make it clearer than the point cloud which encompasses multiple scenarios.

---

> > ### Author Response · Authors · 2022-12-07
> > **Potential remaining concerns**
> >
> > Since the discussion period is almost over, we wanted to make sure that our answers answered your concerns adequately.
> > If not, we would be happy to go into more detail on any outstanding concerns you may have.

---

> > > ### Comment · Reviewer_Sb8P · 2022-12-11
> > > **Reply**
> > >
> > > Sorry for the late reply. Thank you for the detailed answers. However, I am still unsatisfied with the assumptions made in the paper and the impacts that it can bring to the community. So I keep my score and believe the paper needs an entire modification.

---

### Official Review · Reviewer_U7yP · 2022-10-25

**Confidence:** 4
**Correctness:** 2
**Technical Novelty And Significance:** 3
**Empirical Novelty And Significance:** 3
**Recommendation:** 6

**Clarity, Quality, Novelty And Reproducibility:**

Overall, the paper is clear and easy to read. I think this paper makes a useful contribution in terms of reproducibility and demonstrating their proposal. If the authors tweak their claims to better align with the final results, I believe it is a strong submission.
A quick question about Fig 3 and Fig 4:
* Why is the embedding rank always plotted in the x-axis? I thought the performance for representations would be compared to the rank of representations, instead of the embeddings. Did the authors find a better correspondence to the rank of embeddings? Or is this done owing to a different design choice (I might have missed this in the text, apologies for that).

This paper is among the early works to advancing model selection in an SSL pipeline towards a more unsupervised framework. Although there is some concurrent work that aims to solve this problem, this work is still valuable. I believe if the authors can incorporate some of the aforementioned weakensses by taking into light recent work in the field, it would be a very valuable contribution.

**Strength And Weaknesses:**

Strengths:
1. The paper is very well-written and easy to follow and understand.
2. The method is theoretically motivated and provides an insight into why such a metric is useful to observe.
3. The authors carry out extensive experiments to support their proposal. Also, this work could allow other researchers to replicate the experiments (given they have sufficient compute budget, which is indeed a challenging ask) and thereby lead to advancing the state of SSL in vision.

Weakenss:
1. A key concern I have regarding the work is the claim about monotonic relationship between RankMe and performance. A detailed theoretical explanation in the Appendix of Stringer et al. 2019 (https://www.nature.com/articles/s41586-019-1346-5) where they describe how high rank indicates lack of smoothness of the representation manifold, thereby leading to inferior generalization. Indeed, follow up work from Ghosh et al. 2022 (https://arxiv.org/abs/2202.05808) demonstrate a non-monotonic relationship between eigenspectrum decay and linear probe performance. Furthermore, Fig. 3 right plots (corresponding to the performance for representations -- arguably the more common practice in SSL) seems to demonstrate a similar non-monotonic relationship (possibly under-represented due to the log scaling of the x-axis) between rank and performance (clear in VicReg points). Given this non-monotonic relationship, the authors would probably have to do minor tweaks to their model selection proposal and algorithm but it is worth mentioning in the paper for the reader to clearly understand this phenomenon.
2. Another concern I have corresponds to the discrepancy in the language of the claims in the introduction/abstract and the main text. The authors present their method as a surrogate for model performance, although it is only a necessary but not sufficient condition. I would suggest indicating that RankMe has a necessary but not sufficent relationship with model performance in their claims to make the scope and utility clear to the reader.
3. The necessary but not sufficient relationship is not surprising because RankMe is agnostic to the label structure. A similar phenomenon is noted in Ghosh et al. 2022 (https://arxiv.org/abs/2202.05808), where they use eigenspectrum decay as a measure of representation quality.
4.  A minor point, mostly related to a particular claim in the submission, is that RankMe is the first attempt at unsupervised model selection. Although this is concurrent work, but a recent NeurIPS submission aims to achieve similar goals (https://nips.cc/Conferences/2022/Schedule?showEvent=53893). Given the abstract (and the author list), I believe this work is based on the eigenspectrum decay coefficient from Stringer et al. 2019 and I would be curious to know what the authors think about the relationship between RankMe and $\alpha$.

**Summary Of The Paper:**

The authors tackle an interesting and pertinent problem in self-supervised learning, specifically model selection without using task-specific labels. To that end, the authors propose using the rank of the embedding matrix learned in an SSL pipeline as a surrogate metric for model selection. They provide theoretical motivation based on Cover's theorem and Shannon Entropy of the eigenspectrum to use rank as the metric of performance. Given the recent literature on dimensionality collapse, the proposal seems reasonable and is supported by extensive empirical results spanning large-scale datasets and multiple SSL frameworks.
Overall, I think it's an interesting proposal but has some caveats, as described below.

**Summary Of The Review:**

I believe this paper is in the correct direction and has valuable insights that could help the community. I do feel in its current form, there are some gaps that need to be addressed, and thereby I have rated it 5. But if the authors can address some of the aforementioned concerns, I am happy to increase my rating above the acceptance threshold.

---

> ### Author Response · Authors · 2022-11-16
> **Response to review**
>
> > A key concern I have regarding the work is the claim about monotonic relationship between RankMe and performance. A detailed theoretical explanation in the Appendix of Stringer et al. 2019 (https://www.nature.com/articles/s41586-019-1346-5) where they describe how high rank indicates lack of smoothness of the representation manifold, thereby leading to inferior generalization.
>
> We agree with the reviewer’s concerns that high-rank is not sufficient for high representation quality. In fact random embeddings will have maximum rank and arbitrarily poor generalization performances. In fact, we have made sure to emphasize that the rank of a model’s embedding to be high is a necessary but insufficient condition for good downstream performances. This was also observed e.g. with VICReg in which the relation between rank and performance is monotonic only up to a certain point after which the rank of the representation did not correlate well with downstream task performance. We have made sure to emphasize this limitation in sections 3.2 and 4.1.
>
> >Given this non-monotonic relationship, the authors would probably have to do minor tweaks to their model selection proposal and algorithm but it is worth mentioning in the paper for the reader to clearly understand this phenomenon.
>
> We thank the reviewer for this suggestion and we have clarified our claim and limitations throughout the manuscript; we also made sure that all the results were provided either in the main paper or in the appendix to be sure that the limitations of RankMe can be easily assessed.
>
> >A minor point, mostly related to a particular claim in the submission, is that RankMe is the first attempt at unsupervised model selection. Although this is concurrent work, but a recent NeurIPS submission [...]I would be curious to know what the authors think about the relationship between RankMe and alpha-REQ
>
> We thank the reviewer for pointing out this (concurrent) submission to us. Our goals are indeed pretty aligned (to provide a label-free cross-validation method) and although this is concurrent work, we have added thorough comparison with alpha-ReQ in Fig. 1’s table and in appendix E. In that light, we have also toned down our claims (no longer claiming any “first attempt”). For fair comparison, we have added a comparison to alpha-ReQ, both for hyperparameter selection, as well as reproducing our results and verifying how alpha-ReQ works in more complex settings (ImageNet pretrainings in a controlled setting + large scale evaluation.). This detailed analysis is available in supplementary section E.
>
>
> We found that RankMe outperforms alpha-ReQ when looking at performance on the representations, and it especially did not suffer from drops that were as significant as alpha-ReQ. However when evaluating on the embeddings, and thus applying alpha-ReQ on them, the performance is much more similar between alpha-REQ and RankMe. Note however that the power law prior of alpha-ReQ breaks in this setting (see supplementary section E) and must be interpreted carefully.
>
> We also looked at applying alpha-ReQ on the embeddings and evaluation on the representations as for RankMe, and found that in this setting a similar trend as for RankMe can be seen, where performance tends to increase as alpha tends to 0.
>
> > A quick question about Fig 3 and Fig 4: Why is the embedding rank always plotted in the x-axis? I thought the performance for representations would be compared to the rank of representations, instead of the embeddings. Did the authors find a better correspondence to the rank of embeddings?
>
> We found that the rank of representations correlates poorly with downstream performance since they are almost always full rank (for the classical rank operator) or of similar rank (for our entropic measure), due to the collapse happening in the projector. See for example [1] for a more detailed analysis of this phenomenon.
>
> [1] Jing, Li, et al. "Understanding dimensional collapse in contrastive self-supervised learning." arXiv preprint arXiv:2110.09348 (2021).

---

> > ### Comment · Reviewer_U7yP · 2022-12-05
> > **Reply to authors**
> >
> > I would like to thank the authors for putting in the effort to respond to my comments and make significant changes to the manuscript.
> > I believe that the authors agree with my concern about the non-monotonic relationship between rank and performance as well as the necessary but not sufficient relationship. Although I see that the authors have included the point about rank being a necessary condition, it is not very clear to me if they incorporated the point about the non-monotonic relationship.
> >
> > I would also like to commend the efforts put in by the authors in replicating the $\alpha$-ReQ metric and adding large-scale comparisons. This is indeed a great effort and I believe it adds significant insight to the community. Although I don't think it is a big issue in evaluating this work (given that $\alpha$-ReQ is concurrent work), I think the current comparison is slightly at odds with the actual algorithm proposed in [$\alpha$-ReQ page 9](https://openreview.net/forum?id=ii9X4vtZGTZ), whereby the authors do not propose selecting the representation with lowest $\alpha$. Nevertheless, I believe the comparison presented here adds to the understanding of the two methods wrt in-distribution and ood model selection.
> >
> > Keeping these points in mind, I have slightly increased my score.

---

> > > ### Author Response · Authors · 2022-12-07
> > > **Follow up response**
> > >
> > > We would like to thank the reviewer for acknowledging the inclusion of $\alpha$-ReQ, as well as the updates to the manuscript.
> > >
> > > > It is not very clear to me if they incorporated the point about the non-monotonic relationship.
> > >
> > > Thank you for raising this point. While we cannot modify the manuscript anymore we will make sure to incorporate this better.
> > > We agree that the relationship is not always monotonic (even for a given hyperparameter), mostly due to drops in performance near the maximum rank attainable by a given method. It will be interesting to see if and how this non monotonicity (in particular at which point increasing the rank further becomes detrimental) can be captured to further improve existing approaches.
> > >
> > > > I think the current comparison is slightly at odds with the actual algorithm proposed in $\alpha$-ReQ page 9, whereby the authors do not propose selecting the representation with lowest $\alpha$
> > >
> > > To clarify our protocol, in Figure 1 we applied $\alpha$-ReQ by selecting the model with the value of $\alpha$ (computed on the representations) closest to 1, in order to fit closer to the original work. The original algorithm also relied on using linear classification accuracy which goes against our goal, so we adapted the algorithm to fit the original insights (better performance with $\alpha$ close to 1).
> > > For Table 1 where we evaluate after the projector, we applied the same approach but computed $\alpha$ after the projector since this would be the "representations" used for downstream evaluation.
> > >
> > > The goal of Figure S7 was to see if $\alpha$-ReQ can be applied identically as RankMe, differing from the original work significantly. Here we did compute $\alpha$ on the embeddings and the performance on the representations. We did not expand on this comparison due to the significant difference to the original approach, especially since dimensional collapse breaks the power law assumption. We hypothesize that in this setting $\alpha$-ReQ is influenced by collapse (as can be seen in figure S9) which leads to links with performance that are aligned with the goal of RankMe. The target $\alpha = 0$ was not used in our main experiments.

---

### Author Response · Authors · 2022-11-16
**General answer to the reviews**

We thank all the reviewers for their thorough reviews of our submission. We believe that their suggestions have made our submission stronger and we propose a per-reviewer answer to address individual concerns. We also provide this general answer summarizing our contributions, the main comments from the reviewers and the rebuttal’s edits.

**Summary of our contributions:**

We introduced RankMe that enables the selection of Self-Supervised Learning (SSL) hyper-parameters without using any labels on the target nor source task. RankMe computes the Entropy of the normalized singular values of the deep network’s outputs and thus does not require any training or tuning, and works well even if the embeddings are not full-rank. We validate RankMe on

- Architectures: Resnet50, convnext
- Methods: SimCLR, VICReg, DINO, VICRegContrastive, VICRegExp
- Source datasets: Imagenet, INaturalist
- Target datasets: Imagenet, iNaturalist18, SUN397, Places205, StanfordCars, EuroSat  CIFAR10, CIFAR100, PascalVOC07, Food101, CLEVR-count.

Our experiments validating RankMe’s ability to cross-validate hyper-parameters now consist of over **100** model trainings and more than **900** linear model trainings (evaluations).

**Summary of reviewers’ comments:**

All reviewers have appreciated the submission since the need for label-free cross-validation methods in SSL is crucial. All reviewers also greatly appreciated the thorough empirical validation that we conducted. The main pitfalls raised by the reviewers were:
- writing typos and grammatical mistakes; we have corrected those
- lack of comparison with alpha-ReQ; we have added those in Figure 1’s Table and Appendix E and we see that RankMe outperforms alpha-ReQ is most cases
- Need for additional empirical validations: we have added one new SSL method for which RankMe’s cross-validation works great (DINO), five new target datasets, one new source dataset
- RankMe’s limitation for cross-validation between architectures; we have made sure that this limitation is repeatedly mentioned, and we also remind the reviewers that this limitation is also present in previous work (alpha-ReQ) that also aims at providing label-free cross-validation. We thus see this limitation as a rich future research opportunity for any developed method (alpha-ReQ and ours) rather than a limitation of the current method

We hope that our answers and edits (in blue in the submission, except for small typos) along with the per-reviewer answers will address all of the reviewers’ concerns, and we remain available for further discussion.

**Major rebuttal’s edits:**

- Additional target datasets: for all methods (ours and alpha-REQ) we provided additional downstream datasets: CIFAR10, CIFAR100, CLEVR-count, FOOD101, PascalVOC07 validating that RankMe does provide near oracle cross-validation outperforming the existing alpha-ReQ method
- Additional source dataset: INaturalist as source dataset with VICReg and SimCLR (Appendix B): VICReg covariance cross-validation when trained on INat (training and RankMe evaluation) selected an hyper-parameter that outperformance the oracle which was also on INaturalist (which corresponds to cross-validation with labeled INaturalist but deploying model on Imagenet without using its labels), we also report of cross-validations on SimCLR with INaturalist source target
- Additional comparisons: we added comparison with alpha-ReQ in the main table of Fig. 1 where we see that in practical scenarios, RankMe outperforms alpha-ReQ
- Additional SSL method: we added DINO on ImageNet (Appendix C): RankMe works and we either recover the oracle baseline for temperature of the student cross-validation, and we lose 0.1 point when cross-validating the temperature of the teacher

---

### Decision · Program_Chairs · 2023-01-20

**Decision:**

Reject

**Justification For Why Not Higher Score:**

Low reviews, limited novelty, limited applicability.

**Justification For Why Not Lower Score:**

N/A

**Metareview: Summary, Strengths And Weaknesses:**

This paper argues that one can predict or compare the quality of a learned representation from JE-SSL in an unsupervised manner from the embedding matrix rank.   They connect this to theory and show empirically that the rank is predictive of downstream performance across a variety of datasets.

The reviewers found the motivation compelling and the presented approach simple and effective.  They found the paper well written and noted that the empirical evaluation was extensive.  However, the scores were ultimately 3, 3, 6 (raised from 5 during the discussion).  The major concerns of the reviewers were that the proposed strategy seemed limited to hyperparameter selection of JE-SSL, novelty (one reviewer noted that dimensional collapse is a known issue with SSL) and somewhat incorrect claims.  One reviewer found the paper hard to follow and not well organized.  Although this work is promising, the average review score was unfortunately low and none of the reviewers seemed willing to strongly champion the paper.  Therefore, the recommendation is to reject the paper.  Hopefully the reviews provide useful feedback to improve the manuscript for a future submission.

It seems clear that the work has potential and addresses an interesting open question in the literature (i.e. how to compare / select SSL representations in an unsupervised manner).  It seems that a clear direction forward is to explore how this can be used to compare architectures and SSL methods more generally, rather than limiting to hyperparameter selection for a single method.